# Fast, efficient, and accurate neuro-imaging denoising via supervised deep learning

Shivesh Chaudhary [1], Sihoon Moon [1] & Hang Lu [1,2] ✉

Volumetric functional imaging is widely used for recording neuron activities in vivo, but there exist tradeoffs between the quality of the extracted calcium traces, imaging speed, and laser power. While deep-learning methods have recently been applied to denoise images, their applications to downstream analyses, such as recovering high-SNR calcium traces, have been limited. Further, these methods require temporally-sequential pre-registered data acquired at ultrafast rates. Here, we demonstrate a supervised deep-denoising method to circumvent these tradeoffs for several applications, including whole-brain imaging, large-field-of-view imaging in freely moving animals, and recovering complex neurite structures in *C. elegans*. Our framework has 30× smaller memory footprint, and is fast in training and inference (50–70 ms); it is highly accurate and generalizable, and further, trained with only small, non-temporally-sequential, independently-acquired training datasets (~500 pairs of images). We envision that the framework will enable faster and long-term imaging experiments necessary to study neuronal mechanisms of many behaviors.

Fluorescent functional imaging is ubiquitous in neuroscience research in model systems. The persistent goal is to image wider (more cells and larger areas), deeper, and faster, while enhancing signal-to-noise ratio (SNR). For commonly available functional imaging setups such as point-scanning or spinning disk confocal systems, tradeoffs exist between SNR in images and microscopy parameters such as imaging speed (exposure time), field-of-view (FOV), image resolution, length of recording etc. While advancements in genetically encoded calcium and voltage indicators, and new microscopic techniques[1–10] with high spatiotemporal resolution and large FOV have relaxed the requirements and driven the development of whole-brain imaging methods in several organisms, tradeoffs still exist in several model organism systems. For instance, in the nematode *C. elegans*[3,4,11–13], SNR in images is limited due to the requirement of small exposure time to capture neural dynamics at 3–6 volumes/s and to prevent motion artifacts. While SNR can be improved by increasing laser power, this leads to photo-bleaching of fluorophores and photo-toxicity, thus limiting the length of recordings, especially during longer timescale behaviors. Additionally, to image neurons across the whole animal, FOV must be

further expanded, requiring lower magnification, higher laser power, and exacerbating photobleaching. Advanced microscopy techniques can help mitigate these tradeoffs[9], but require specialized expertize to implement and maintain the necessary hardware, making them inaccessible to many researchers.

Recently deep learning enhanced microscopic techniques[14–16] have been developed that significantly overcome the tradeoff between imaging speed and SNR in images. However, these techniques either require expertize in characterizing the microscopy system at hand for generating realistic training data, such as the axial light propagation[14] or they require light-field microscopy setups[15,16] that are not commonly available to all researchers. Further, whether these methods can perform at low laser power conditions that are critical to prevent photobleaching and enable long-term recording of neuron activities is not currently shown. Thus, an orthogonal method to enhance SNR, to circumvent the tradeoffs, would be enabling in many studies.

An alternative strategy that has been established recently and has achieved state-of-the-art results to overcome tradeoffs in microscopy is deep-learning-based image denoising[17–26]. In these methods, a deep

[1]School of Chemical & Biomolecular Engineering, Georgia Institute of Technology, Atlanta, GA, USA. [2]Petit Institute for Bioengineering and Bioscience, Georgia Institute of Technology, Atlanta, GA, USA. ✉e-mail: hang.lu@gatech.edu

neural network is trained to recover high SNR fluorescent images from low SNR images acquired with low exposure time or low laser power conditions. These include supervised[17–22] and unsupervised[23,24,26,27] methods. Unsupervised methods offer the benefit of training on the data to be denoised itself thus no training data collection is needed. Despite the success of denoising methods, their application on downstream analyses such as high SNR calcium trace extraction from videos has been shown in only a few model organisms and microscopic techniques, all using unsupervised methods. For instance, DeepInterpolation[27] and DeepCAD[26], demonstrate high-quality calcium trace extraction on 2D two-photon imaging data in mice. While impressive and without requiring curated ground truth, these methods do require large training data sets (~100,000 frames for DeepInterpolation and 3500 frames for DeepCAD); further, pre-registration of the videos (or videos with minor movements) are required before training, which also necessitates ultrafast imaging rates. DeepCAD also shows decreasing accuracy for data acquired at slower imaging rates, demonstrating that information in temporally linked images is important for denoising. Practically, these models also have a large memory requirement for training and inference. While new advances in microscopy greatly improve imaging speed and FOV[10], generating such large-scale ultrafast recordings for 3D imaging in models organisms is currently not feasible for all researchers with commonly available confocal systems. Additionally, training these methods on calcium activity recordings in moving animals would require a non-trivial pre-registration step, and training results would be contingent on the accuracy of the registration step.

Compared to unsupervised methods, supervised methods for image processing are expected to achieve higher denoising accuracy and be more generalizable. Currently, supervised methods have not been used for video data denoising and extracting calcium traces. This is likely due to several factors. For instance, if supervised methods are to be trained using temporally linked data, akin to unsupervised methods, custom microscope setups will be needed that can collect low and high SNR video data simultaneously. In contrast, if supervised methods are to be trained with non-temporally linked data, it is not immediately apparent whether the temporal structural features in the dynamical data (as in calcium imaging experiments) can be preserved from independently denoised images. It is also not obvious to what extent the supervised models can be generalized. The wide deployment of these models will also be dependent on several practicalities such as model size, inference speed, and memory requirement on the computation. Here we show that supervised deep denoising can achieve high accuracy in extracting high-SNR calcium traces from noisy videos. Our optimized models are 20–30× smaller in memory footprint, 3–4× faster in inference speeds, and can be trained with as few as 500 pairs of images that are temporally independent and collected across different samples. With the use of temporally independent data for training, fast imaging rate for training data collection and pre-registeration step are not required; further, networks can be trained with a variety of images across animals with different posture configurations, neuron morphologies, cell labeling techniques (soma, membrane etc.) and markers (RFP, GCaMP etc.), thus improving the generalizability across conditions and noises.

## Results

### Optimized deep neural networks for denoising images

To address the challenges of extracting clean calcium traces from noisy calcium imaging videos in common applications, we designed Neuro-Imaging Denoising via Deep Learning (NIDDL), a convolutional neural network (CNN) pipeline that can be trained using only a small set of training non-video data (Fig. 1). The ability to work with independently acquired image training pairs (not from videos) greatly improves the generalizability and ease-of-access because of the much relaxed requirements in data acquisition. For instance, to obtain the

ground truth for training, images can be acquired for immobilized samples, with little photobleaching (by using independent samples), at different times, and possibly across different biological conditions (e.g., different strains). This enables more researchers using a wider set of instruments and in wider range of biological settings to denoise neural images and recordings. The pipeline takes in independent pairs of noisy (acquired either with low laser-power or short exposure-time) and high SNR image stacks, acquired across samples and reagents (Fig. 1A). Subsequently, efficient denoising CNNs are trained using the non-video data. In the application phase, trained networks are applied to denoise video data by independently denoising each volume in the video. Finally, high quality calcium traces are extracted from the denoised video using a conventional calcium signal extraction pipeline in *C. elegans* that involves cell segmentation, cell tracking, and signal extraction (Fig. 1A). As an example, microscopy conditions used for whole-brain calcium activity recordings lead to significant loss of SNR in images (Fig. 1B), which makes densely packed nuclei in images barely distinguishable (Supplementary Fig. 1). Low SNR in images can significantly reduce the accuracy of intermediary tasks such cell segmentation and tracking, thus making downstream analysis of neuron activity data extremely slow and challenging. We demonstrated that trained networks can significantly recover nuclei structure from these noisy images (Fig. 1B and Supplementary Fig. 1).

To achieve a fast and data-efficient CNN with a small memory footprint, we optimized several network hyper-parameters ("Methods" – 'Network Optimization'). For instance, starting with vanilla UNet[17,28] and Hourglass architectures[29], we tested several design choices such as kernel size, channel depth, depth of architecture, and presence or absence of residual connections (Supplementary Figs. 2, 3). Additionally, we compared architectures across L2 and L1 loss functions used commonly in image restoration tasks[17,20] (Supplementary Fig. 4), and three different training modes (Supplementary Fig. 5) including 2D mode, 2.5D mode, and 3D mode ("Methods" section). The optimal models significantly reduce the number of parameters and memory footprint by fixing channel depth across all layers. This allows networks to (1) be deeper, i.e., have more convolutional blocks compared to CARE with default parameters[17], and (2) use residual connections within each convolutional block that are not present in default UNet. Compared to previously established methods such as CARE[17], RCAN[19], and default UNet and Hourglass architectures, our optimized architectures are 20–30× smaller in memory footprint, have 3–5× faster inference time, are 2–3× faster in training (Fig. 1B, C). We show that for whole-brain imaging applications, model accuracy plateaus at training with 500–600 image pairs (corresponding to 25–40 pairs of whole-brain stacks) (Fig. 1D and Supplementary Fig. 6), which is much smaller than the number of images used for training in recent methods DeepCAD (3500 frames) and DeepInterpolation (~100,000 frames). Thus, networks can be easily trained in individual labs specific to individual experimental and instrumentation conditions.

We have also tested CNNs trained with L2 or L1 loss and show that they achieve similar accuracy (Supplementary Fig. 4), with L1 loss training being more stable across different instances of training. Further, we noticed that L1 loss performs better in RMSE and PSNR metrics whereas L2 loss performs better in SSIM metric. This could be because L1 loss is more suitable to handle the type of noise present in experimental data whereas L2 loss is more suitable to preserve structural information. Finally, we tested three modes of training that differ in 3D spatial context used by networks for denoising (Supplementary Fig. 5). These modes include (1) 2D mode where input and output to the networks are 2D images, (2) 2.5D mode where input to the network is a 3D stack consisting of *z*-planes above and below the image to be denoised and output is the middle denoised 2D image, and (3) full 3D mode where input to the networks is 3D stack and output is also 3D stack. Comparisons show that training with 2D images, rather than

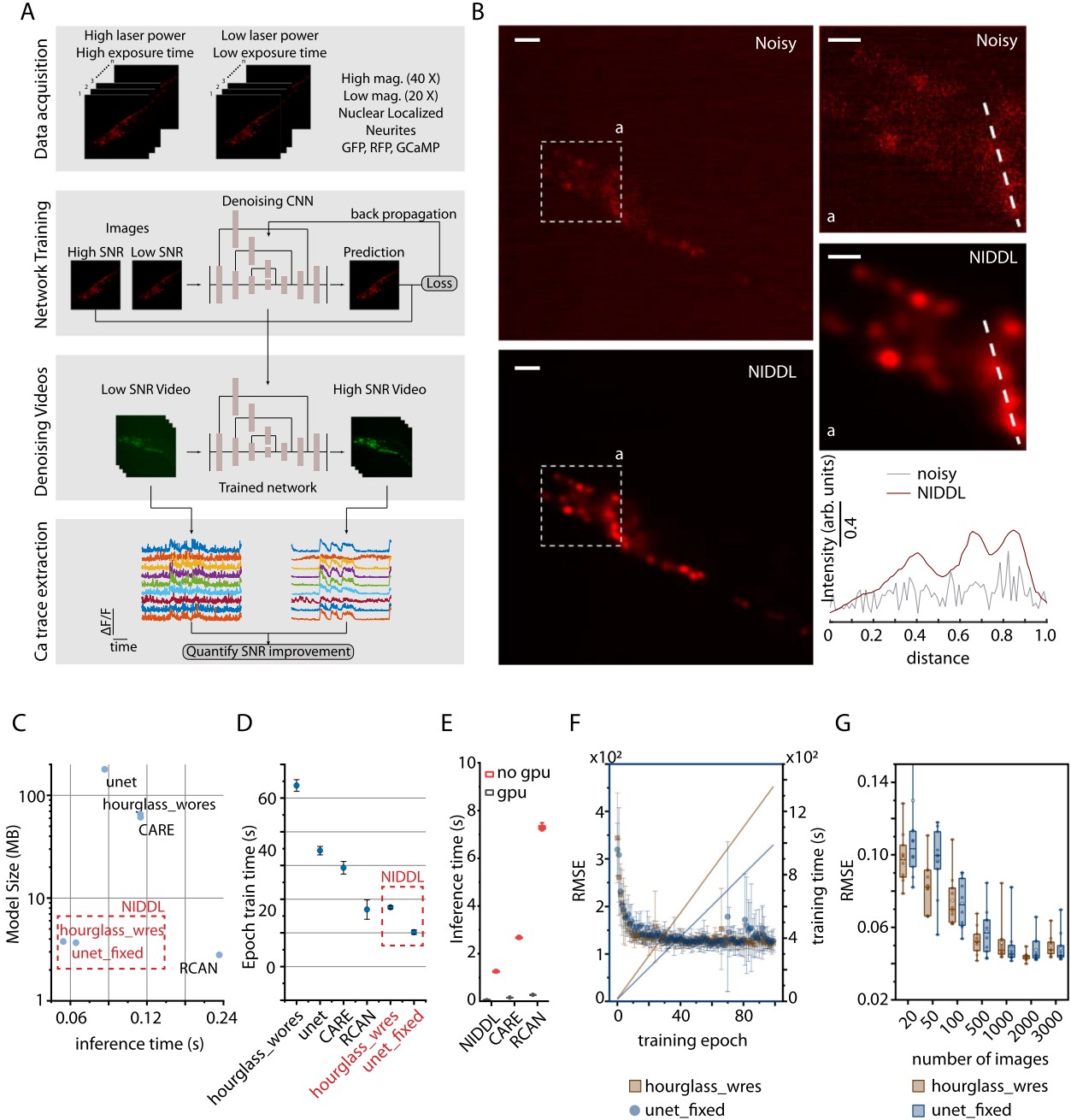

**Fig. 1 | Neuro-Image Denoising with Deep Learning (NIDDL) framework.**
**A** Overview of the SL framework to recover high SNR from a variety of imaging conditions. **B** An example noisy image (1 zplane from 3D stack, strain OH16230) and corresponding deep-denoised image (scale bar 10 μm). Top-right panels show insets 'a' in noisy and denoised images (scale bar 5 μm). Bottom-right panel shows intensity along dotted lines in noisy and deep denoised images. **C** The optimized neural network architectures 'unet_fixed' and 'hourglass_wres' have 20–30× lower model size (3.77 MB and 3.66 MB) and 3–4× faster inference time (average 48.9 ms and 68.7 ms per 512×512 image calculated across 600 images), compared to CARE, RCAN, and non-optimized UNet, Hourglass. **D** Per-epoch training-time comparison across neural network architecture variants (batch size of 50 images, epoch training size of 1000). Each dot corresponds to average epoch train time across 100 epochs for each instance of trained networks. Error bars indicate standard deviation across 5–10 instances of training with random subset of total data used for training each instance. **E** Denoising-time comparison of deep-learning methods when inference is

performed with and without GPU. ($n$ = 50–600 images). Box center indicates median, edges 25th and 75th percentile, and whiskers 5th and 95th percentile.
**F** Training curves for the optimized neural network architectures and cumulative epoch training-time with batch size of 50 and epoch training size of 1000 images. Each dot in training curves corresponds to average RMSE loss across 10 instances of training. Error bars correspond to standard deviation across 10 instances of training. **G** Accuracy-vs-training-data-size tradeoff for optimized architectures. Each dot corresponds to mean RMSE accuracy on 600 test images for one instance of trained network. Ten instances were trained for each condition with random subsets of training data. RMSE accuracy plateaus above 500 images for both architectures. Data comes from strain ZIM504. Box center indicates median, box edges indicate 25th and 75th percentile, whiskers indicate 5th and 95th percentile. Architectures highlighted in red in **C**, **D** correspond to NIDDL. Source data for **C**, **F**, **G** are provided in Source Data file.

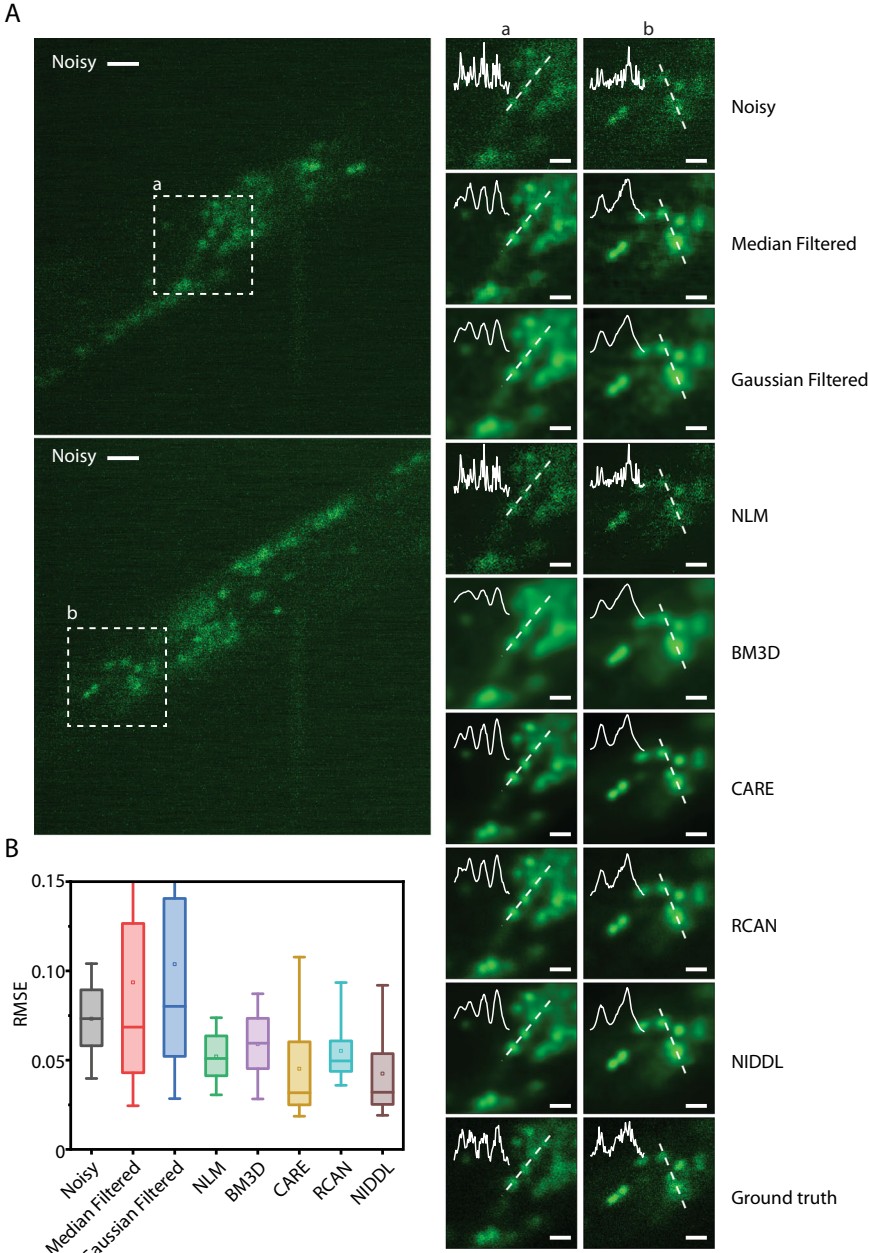

**Fig. 2 | NIDDL denoises whole-brain images in *C. elegans*. A** (Left) Two example noisy images (single *z* planes) from noisy whole-brain image stacks (acquired at low laser power) (scale bar − 10 μm). (Right) corresponding denoised output generated by different methods shown for the dotted box in noisy images (scale bar − 5 μm). Cell nuclei are labeled with nuclear localized GCaMP5K. Data come from strain ZIM504. Inset shows intensity profile along the dotted line. **B** Comparison of RMSE to ground-truth high SNR image across noisy images, and denoised images output by various methods including Median filter, Gaussian filter, NLM, BM3D, CARE, RCAN, and NIDDL (*n* = 600 images). Box center indicates median, box edges indicate 25th and 75th percentile, and whiskers indicate 5th and 95th percentile. Source data are provided in Source Data file.

3D stacks, is sufficient (Supplementary Fig. 5), possibly because more training data is needed for 3D mode of training. Practically, 2D images can be acquired easily using commonly available setups thus simplifying the training step. Importantly, these memory-efficient and fast models can be used widely without expensive GPUs. When comparing inference time of models without GPUs, NIDDL achieves an average inference time of 1.25 s whereas CARE and RCAN denoise images in much longer time of 2.67 and 7.29 s respectively (Fig. 1E).

To characterize the performance of NIDDL, we worked with *C. elegans* strains with whole-brain neuronal labels. We took advantage of microfluidic immobilization of animals to avoid the complex image pre-registration step across image pairs before training the networks, and to acquire data in high-throughput manner[30]. Acquired pairs of non-sequential image data across samples are used as input to train the CNN. Trained networks are then applied to noisy video frames independently to recover clean images. Subsequently high SNR calcium traces are extracted from denoised videos. We show that deep denoising recovers structures in noisy whole-brain images with clear distinction of nuclei (Fig. 2, Supplementary Fig. 1, 7 and Supplementary Movie 1), which can greatly improve the nuclei segmentation performance[31], and thus the accuracy of downstream tasks such as cell identity annotation[32]. We bench-marked NIDDL's performance with those from other approaches ("Methods" – 'Comparison against other methods'). Representative images show that NIDDL produces cleaner denoised images closer to the ground truth images, while simple denoising methods such as Median and Gaussian filtering, as well as

advanced non-deep learning based methods such as NLM and BM3D, suffer from either blurring artifact or not recovering information (Supplementary Fig. 7). Quantitatively, the optimized NIDDL model achieves high accuracy on held-out datasets, outperforming traditional denoising methods, non-deep learning based methods such as NLM and BM3D, and deep learning methods such as RCAN[19] (Fig. 2B and Supplementary Fig. 8). While the recently published algorithm CARE produces similar accuracy as NIDDL, the advantage of NIDDL is smaller model size and real-time inference speed (Fig. 1C), which would be important for applications that would require near real-time feedback, e.g., closed-loop optogenetic interventions.

To test the generalizability of the approach, we trained separate network instances on data collected across a variety of conditions and compared within-condition accuracy with across-condition accuracy. These include two whole-brain imaging strains with different levels of fluorophore expression labeling all cells, three levels of laser powers, and three independent experiments on different days. Models trained on independent experiments and strains are particularly generalizable across conditions (Fig. 3A–D). As an example, denoised images output by networks, when networks were either trained on the same strain or a different strain, visually appear indistinguishable (Fig. 3A, B). In both cases, networks significantly recover distinguishable nuclei structure from noisy images (Fig. 3B). When accuracies are characterized, cross-strain model performance also appears similar to that of within-strain models (Fig. 3C). Our results do show, however, that models are sensitive to image-acquisition laser power (Fig. 3D and Supplementary Fig. 10). In comparison, models generalize with a high degree of accuracy across independent experiments (Fig. 3E and Supplementary Fig. 9). In parallel, we conducted an in silico experiment to characterize the robustness of the optimized CNNs against noise levels; we generated realistic 3D synthetic data with densely packed nuclei (Methods – Synthetic whole-brain data generation) across a range of signal levels (photon counts), corrupted by Poisson shot noise and Gaussian readout noise. We show that NIDDL consistently and efficiently denoise the images, better than traditional methods (Supplementary Fig. 11). We hypothesized that as long as a minimum requirement for SNR is met, NIDDL can produce efficient denoising, and that the corruption of the signal by noise beyond a certain threshold cannot be rescued by denoising. Indeed, this notion is corroborated by the characterizations of the SNR in the actual experiments (Fig. 3F) where the SNR levels across laser powers vary vastly, those across strains vary less, and across independent experiments, sessions have similar SNR levels. These results demonstrate that as long as the imaging experiments meet a minimum SNR threshold (~20), NIDDL can efficiently denoise. This points to the advantages of NIDDL, where training data sets can be gathered in a distributed manner and from varied conditions (including from different strains), which would greatly lower the barriers for use in practice.

## High SNR calcium trace recovery using NIDDL

While denoising images in itself can improve the accuracy of many tasks in whole-brain imaging, including segmentation, tracking, and identification, the critical goal is to extract clean calcium traces. We next denoised a whole-brain video ("Methods" – 'Calcium imaging data collection') that was held out from the training (Fig. 4A and Supplementary Movie 2) and extracted traces ("Methods" – 'Denoising and extracting calcium traces'). We note that methods used for calcium signal extraction from two-photon recordings of spiking neurons[33–35] differ from standard methods used for *C. elegans*[11,36–38]. The deep denoised video provides much cleaner traces compared to the original noisy video (Fig. 4B) and correlated neuron activity is detectable visually. Since NIDDL is trained using non-video data, denoising each frame of video independently could introduce artifacts in calcium traces. To establish that NIDDL recovered calcium traces do not contain artifacts, we compared the traces extracted

from denoised video to traces extracted from high-SNR ground-truth video for the same recording. Denoised traces show the same temporal structure in neuron activity as present in high SNR video, thus, denoising does not introduce artifacts (Fig. 4B). Furthermore, denoised traces show much lower mean absolute error (Fig. 4C) and higher correlation to the traces from the ground-truth low-noise video (Fig. 4D). This demonstrates that denoising by NIDDL greatly improves SNR in the frames independently. Further and perhaps more importantly, denoising with NIDDL recovers correlational structure among neuron activities (Fig. 4E), crucial for downstream analyses and interpretation such as PCA based latent activity recovery[11] commonly used in whole-brain data analysis pipelines[39]. We further tested the robustness of NIDDL against different noise levels by denoising and extracting traces from semi-synthetic videos across a range of SNR levels ("Methods" – 'Semi synthetic video data generation'). Deep denoising significantly removes noise from traces (Supplementary Fig. 12A, D) and performs better than traditional methods across all SNR levels (Supplementary Fig. 12E). Lastly, we show that NIDDL denoised traces significantly improve the performance of PCA analysis commonly used for analyzing whole-brain recording datasets[11,40]. Neural activity trajectory in low dimensional space show smooth dynamics in ground-truth video; however, such structure is lost in noisy video (Supplementary Fig. 12B). The NIDDL denoised video successfully recovers the smooth dynamics (Supplementary Fig. 12B) by recovering the correlational structure among neuron activities (Supplementary Fig. 12C). Taken together, these results demonstrate that denoising using NIDDL requires a small set of training data, is forgiving in many experimental constraints, and yet provides excellent performance in accuracy, robustness, and generalizability while using small inference time potentially enabling on-line feedback manipulations from calcium dynamics.

Next, we sought to demonstrate denoising on large FOV data acquired at low magnification ("Methods" – 'Calcium imaging data collection'). The advantage of a large FOV is to capture more cells simultaneously. The challenge with a large FOV recording, however, is low spatial resolution so that each cell corresponds to only a few pixels, and this necessitates higher laser power to boost SNR. Here, we imaged simultaneously many ventral cord (VC) motor neurons in *C. elegans*. To avoid photo-bleaching, we also used low laser power, which results in worse SNR as compared to imaging at 40× (higher NA) (Fig. 5A and Supplementary Fig. 13). We trained NIDDL with temporally independent (i.e., non-video data) pairs of low and high-laser-power images of ventral cord neurons expressing GCaMP. NIDDL was able to remove much of the noise, enabling the detection of cells barely noticeable in noisy images (Fig. 5A and Supplementary Fig. 13). Quantitative comparisons show that NIDDL significantly outperforms traditional denoising methods and advanced non-deep learning based methods (Fig. 5A, B and Supplementary Figs. 14, 15) and achieves similar accuracy to CARE and RCAN. Next, we denoised low-SNR videos held out from training and extracted calcium traces from them. Again, NIDDL enables extraction of high-quality calcium traces from noisy videos, making it much easier to detect coordinated neuron activities (Fig. 5C, Supplementary Fig. 16, and Supplementary Movie 3) barely visible in traces extracted from noisy videos. Importantly, while averaging of cell ROI pixels to extract traces leads to significant SNR improvement in traces from noisy video, it was not enough to recover neuron activity transients and many bouts of activities were lost (Fig. 5C and Supplementary Fig. 16). In contrast, single pixel traces extracted from the NIDDL denoised video recovered all of these transients with much higher SNR and performed just as well as when ROI averaging was used to extract traces from NIDDL denoised video (Fig. 5C and Supplementary Fig. 16). Thus, single pixel trace extraction is enough for NIDDL denoised videos. This demonstration suggests that NIDDL is a truly enabling tool for large FOV applications where SNR levels in images are very low, and each cell corresponds to only a

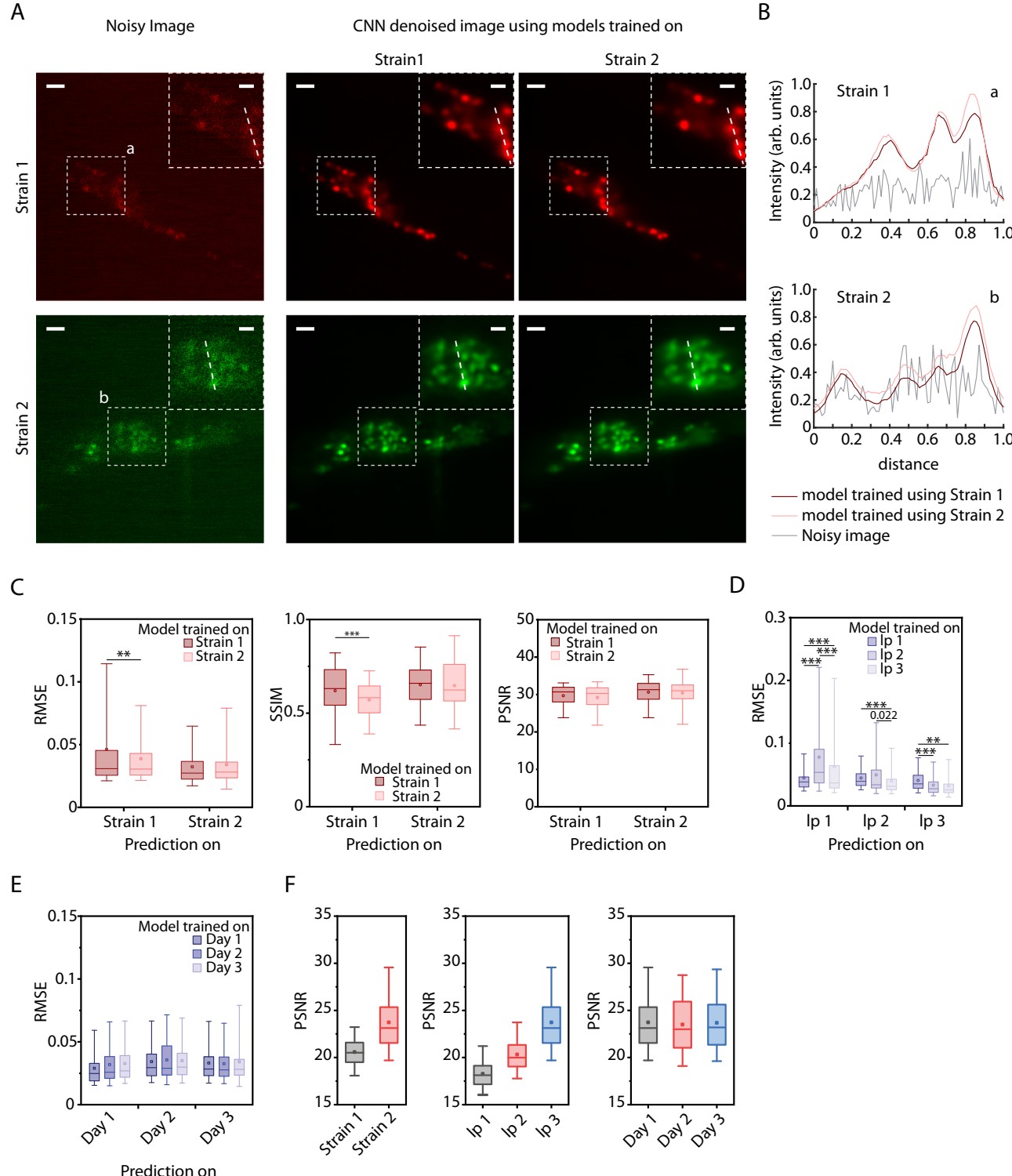

few pixels in images. Further, NIDDL can help avoid photobleaching by enabling imaging at low laser power conditions.

While a simple use of the large FOV and deep denoising is to increase the number of cells observed simultaneously and increase the throughput of experiments by enabling imaging multiple animals simultaneously (Fig. 5C), the technique is truly enabling for imaging moving samples where low exposure time (to reduce blurring type motion artifact) is critical (Fig. 5D). Conventionally for freely moving animals, neural activities are imaged at high magnification fluorescence channel, while behavior is tracked with a second low-

magnification light path. Here, with large-FOV low-magnification imaging, fast exposure times, and deep denoising, animals can be tracked directly on the fluorescence light path, while NIDDL can extract clean calcium traces from these videos, with more cells, without compromising the imaging quality. We demonstrate this by imaging motor neurons' along the ventral nerve cord of freely moving animals. Deep denoising by NIDDL significantly removes noise from calcium traces, resulting in clear bouts of neural activities (Fig. 5D and Supplementary Movie 4). Next, we correlated activities of motor neurons to local body curvature of the animal as it roams. Motor neuron activities recovered

**Fig. 3 | NIDDL performance generalizes across strains and experiments.**
**A** Denoising example on images from 2 strains (OH16230 and ZIM504) with different cell markers when model is trained with strain-specific data. (Left) Example max-projection noisy images from 2 strains; (right) corresponding denoised images by 2 different networks trained on strain-specific data. Scale bar is 10 µm.
**B** Intensity profiles along the dotted lines shown in insets in A for noisy images and denoised images output by 3 different networks. **C** Comparison of strain-specific model performance and across-strain model performance. (Left) RMSE accuracy, (Middle) SSIM accuracy, and (Right) PSNR accuracy on noisy images from 2 strains when networks are trained on specific strain's data. ($n = 100, 3006, 1403, 50$ images for 4 conditions, $**p < 0.01$, $***p < 0.001$, two-sided Holm-Bonferroni paired comparison test). **D** Denoising accuracy on images acquired with three different laser power settings (lp1 – extremely low laser power, lp2 – very low laser power, lp3 –

low laser power) when model is trained with specific laser power setting's data only. Data comes from strain ZIM504. ($n = 88, 2728, 2728, 2262, 100, 2262, 2806, 2806, 226$ images for 9 conditions, $***p < 0.001$, $**p < 0.01$, two-sided Holm−Bonferroni paired comparison test). **E** Denoising accuracy on images from three different days' imaging sessions when model is trained with specific day's data only. Data comes from ZIM504. ($n = 40, 1043, 1043, 1532, 50, 1532, 1403, 1403, 50$ images for 9 conditions). **F** Comparisons of SNR levels in noisy images across conditions. Left – across different strains ($n = 3006, 1403$ images), middle – across different laser powers ($n = 1403, 1364, 1101$ images), right – across different days ($n = 1403, 1532, 1043$ images). Source data for **C**–**F** are provided in Source Data file. In **C**–**F**, Box center indicates median, box edges indicate 25th and 75th percentile, and whiskers indicate 5th and 95th percentile.

by NIDDL showed enhanced correlation to animal curvature (Supplementary Fig. 17B–D) compared to traces extracted from noisy videos. Thus, NIDDL enables recordings where samples move significantly across frames by enabling imaging using low exposure time conditions. By requiring only low light, this approach will also enable more prevalent longer-term imaging with behavior.

### Complex neurite structure recovery with NIDDL

Another application of deep denoising is in imaging subcellular features such as the dendritic processes, which are typically dim and difficult to quantify compared to imaging the soma. Because denoising neurites presents different challenges, we sought to optimize network hyper-parameters specifically for neurites (Supplementary Fig. 18) and chose L2 loss over L1 loss due to slightly better performance. The optimized network recovers structures of neurites from noisy images (Fig. 6A, Supplementary Figs. 18, 19, and Supplementary Movie 5) showing distinct processes barely visible in noisy images. Further, NIDDL enables quantitative characterization of neurite morphology as recovered neurite structure significantly improves neurite segmentation performance using simple methods ("Methods" – 'Neurite segmentation') (Fig. 6D and Supplementary Fig. 20). Compared to non-deep learning based methods, NIDDL again performed better on accuracy (Fig. 6B and Supplementary Fig. 21). Further, NIDDL achieved similar accuracy compared to previous deep learning based methods. To test generalizability across neurite morphology, we tested the performance across two strains labeling neurons with distinct structures (the gentle touch neurons ALM, AVM, and PLM, and the multimodal sensory neuron PVD in *C. elegans*). Models trained only on one strain's data achieved equivalent accuracy across other strain (Fig. 6C and Supplementary Fig. 23). We envision NIDDL being applied to study calcium signal distribution in complex morphologies of mechanosensory neurons.

## Discussion

In this work, we present an easy-to-train, fast, data-efficient, and generalizable deep-learning framework for denoising calcium activity volumetric recordings. While our method has similarities to recently developed supervised learning methods for restoring images[17,20], applications of supervised methods for extracting calcium traces from volumetric recordings in model organisms have not been shown. Here, we demonstrate the utility of supervised denoising methods for various calcium imaging application in *C. elegans*, and highlight key advantages over previous methods, which make them attractive for researchers to adopt easily. First, we demonstrate that networks trained with temporally independent (non-video) data collected across animals, strains, and imaging conditions can be used to recover high-quality calcium traces from video data, thus providing several experimental simplifications. For instance, ultrafast imaging rates for training data collection are avoided, thus enabling more labs to collect data with commonly available microscopy setups. Additionally, since networks are trained with non-video data, complex pre-registration of

images before training is circumvented, making the method suitable for motile animals, such as *C. elegans*.

Second, we demonstrate that networks can be trained with order of magnitude smaller training data (∼500 pairs of images) compared to previous methods DeepInterpolation and DeepCAD. Temporally sequential data used in these previous methods closely resemble the approach of Noise2Noise[23]. This is because consecutive images in pre-registered data from ultrafast recordings can be thought of as coming from one sample with independent noise in each image such that the expectation matches the noise-free sample image. Due to the need of multiple images of each sample to accurately approximate the expectation, the size of the temporal window used in DeepInterpolation and DeepCAD is on the order of ∼70−300 frames; thus, these methods tend to require large amounts of training data. In comparison, supervised learning methods, can be trained with only two images of each sample i.e., low SNR and high SNR images, and do not make assumptions on noises in the data. Thus, supervised methods are advantageous for dynamic data, such as those from (slow) volumetric functional imaging where consecutive frames may not have correlated signals, and those from moving samples where frames are not already registered; furthermore, supervised methods can also be trained with much smaller training data.

Third, we demonstrate the generalizability of networks trained in supervised manner across different strains and experimental sessions. This is possible because the models are trained with pairs of high SNR and low SNR images across a variety of conditions, animals, strains etc., which capture the distribution of SNR levels across experimental conditions that researchers may expect under typical experimental conditions. Thus, these supervised methods can achieve higher denoising accuracy compared to unsupervised methods, making them suitable for high quality calcium signal from new experimental recordings without retraining the networks. In contrast, unsupervised methods trained on one functional recording dataset may need to be trained again for every new video. Furthermore, to eliminate complex pre-registration of images in moving animal recordings during training, supervised training only needs images of stationary samples across various conditions, and trained networks can be applied to images in videos independently to recover high SNR traces. Thus, the pipeline is much more accessible to routine use in calcium imaging in a wide range of scenarios, e.g., in sensory behavior, mating behavior, and social behavior.

Finally, we have optimized networks extensively to significantly reduce the memory footprint and inference time compared to previous deep learning methods. Our current models can achieve real-time denoising speeds making them suitable for experiments requiring real-time feedback such as optogenetic perturbations. Our demonstrations on variety of data sets that include high-magnification whole-brain calcium recordings, low-magnification large FOV calcium recordings, and recovering complex neurite morphology highlight the utility of deep learning based denoising methods. We imagine that our demonstration of deep learning methods for functional imaging

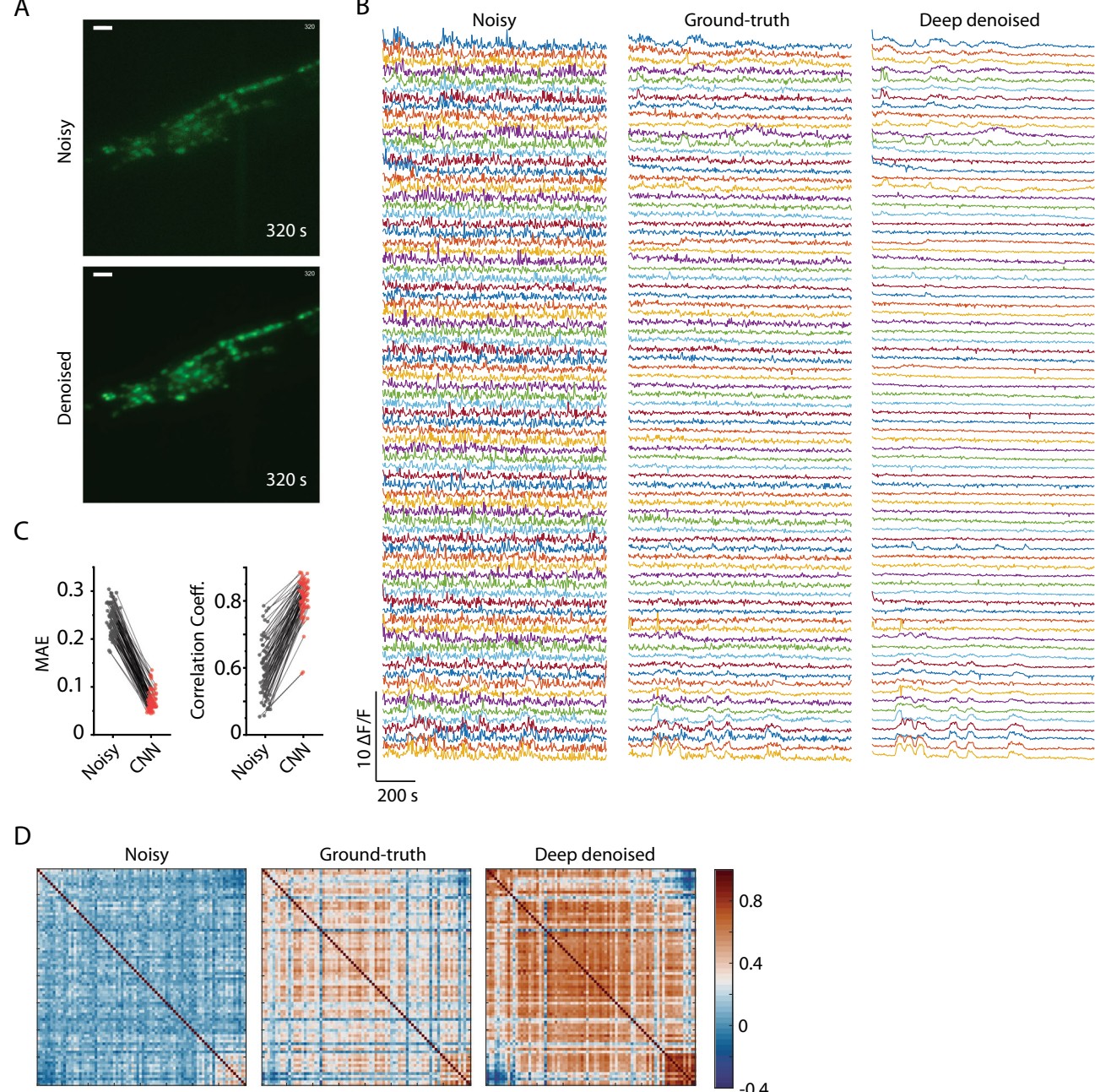

**Fig. 4 | High quality whole-brain Ca trace recovery using NIDDL. A** Top – max projection of an example image stack from a noisy whole-brain video recording (acquired at low laser power). Bottom – corresponding deep denoised output. Cell nuclei are labeled with nuclear localized GCaMP5K. Data comes from strain ZIM504 (scale bar − 10 μm). **B** Neuron activity traces extracted from the noisy video (shown in **A**), high SNR ground-truth video for the same recording (acquired at high laser power), and deep denoised video output by network trained only on separate image data. **C** Cell-wise comparison of mean absolute errors (MAE) (Left) and Pearson correlation coefficients (Right) of traces extracted from noisy video and denoised video to corresponding traces extracted ground-truth video. $n = 80$ cells. Source data are provided in Source Data file. **D** Pairwise Pearson correlation among neuron activity traces extracted from noisy video, ground-truth video, and deep denoised video. Rows and columns correspond to 80 cells.

denoising in *C. elegans* will inspire newer experiments in other model systems such as hydra[41–43], Drosophila[44–50], and zebrafish[39,51–55], where long-term whole-brain and functional recordings are needed to uncover neuronal basis of behaviors that evolve over long time scales[56–58]. NIDDL facilitates such recordings by use of lower laser power and shorter exposure time. Combining this technology with microscopy techniques requiring low light dosage e.g., light sheet[9,51,59,60], or other microscopy techniques such as virtual refocusing[14], light-field reconstruction[15,16], and multiphoton imaging[61]

will enable newer imaging paradigms, recordings of longer durations, and faster frame rates previously not possible.

## Methods

### *C. elegans* culture

For all experiments, animals were cultured using standard techniques[62]. A detailed list of strains used in this work is provided in Table 1. All data are collected using Larval stage 4 (L4) *C. elegans* hermaphrodites.

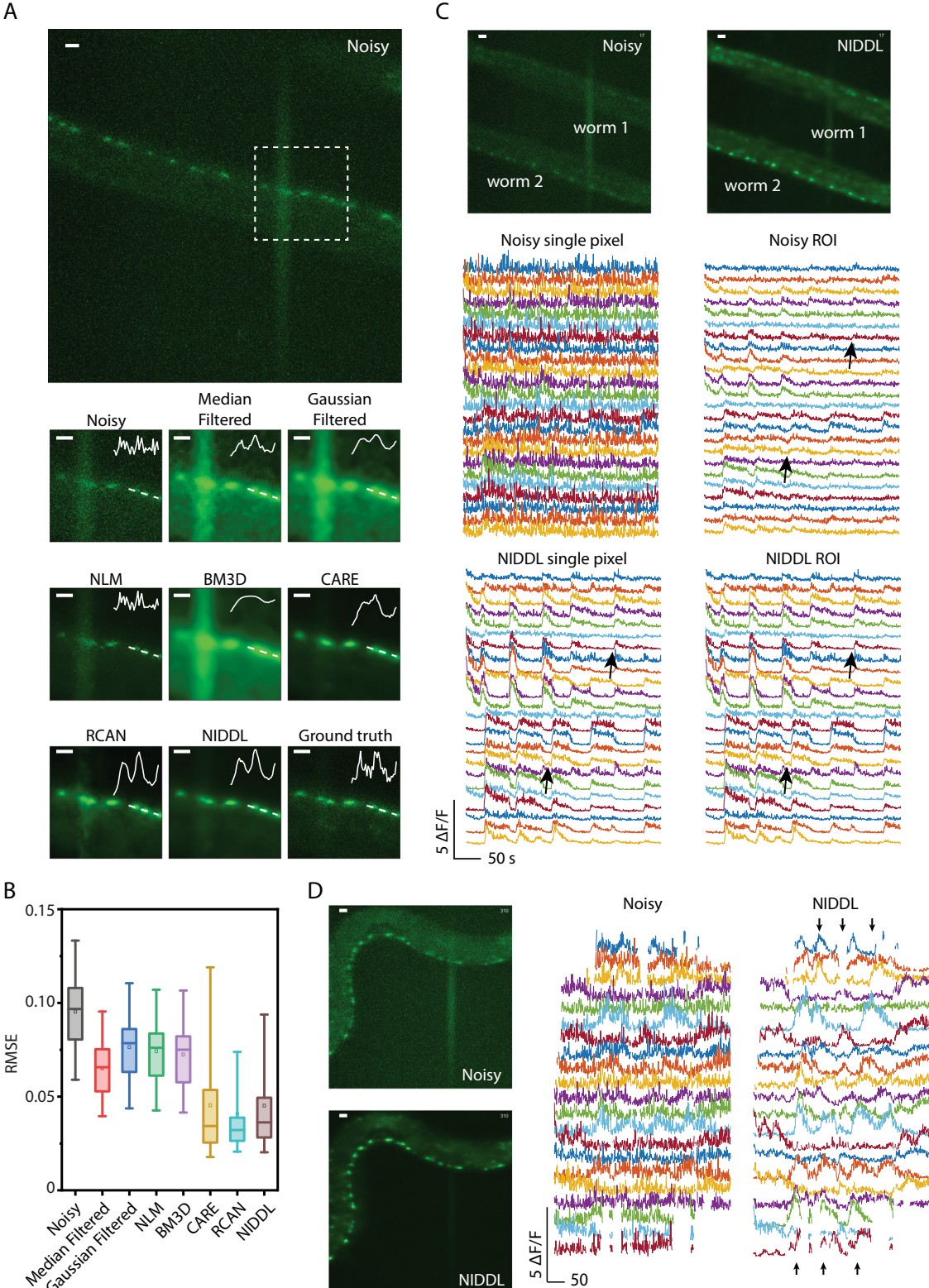

## Training data collection

All imaging was performed using Bruker Opterra II Swept field confocal microscope, with an EMCCD camera. Objective lenses used for each type of data and other imaging parameters are described below.

1. Whole-brain data - Whole brain data was collected using ZIM504 and OH16230 strains. Animals were synchronized to L4 stage and were immobilized in a microfluidic array device to prevent motion. Two 3D stacks (25−30 z planes with 1 μm spacing) were acquired for each animal, one at low laser power and one at the highest laser power setting available in microscope, 10 ms exposure time, and Plan Fluor 0.75 NA 40× air objective. Low laser power image specifies the noisy (low SNR) image and high laser power image specifies the clean (high SNR) image. Neural networks were trained to predict high SNR image from low SNR image as described in the section 'Network Training'. To quantify prediction generalizability across days, independent datasets

**Fig. 5 | High quality Ca trace recovery in large field-of-view imaging of spatially distributed motor neurons. A** (Top) max projection of a large FOV noisy image stack (low laser power) showing ventral cord motor neurons (scale bar − 10 μm). (Bottom) denoised outputs generated for square box marked in the top image (scale bar − 5 μm). Inset shows intensity profile along the dotted line. Cell nuclei are labeled with nuclear-localized GCaMP6s. **B** Comparison of RMSE to ground-truth high-SNR images across noisy images, and denoised images from Median filter, Gaussian filter, NLM, BM3D, CARE, RCAN, and NIDDL (n = 60, 60, 60, 60, 60, 60, 72, and 217 images correspondingly). Box center indicates median, box edges 25th and 75th percentile, and whiskers 5th and 95th percentile. Source data are provided in Source Data file. **C** (Top left) input noisy (low laser power) maximum projection from a large FOV showing ventral cord motor neurons of two animals restrained in microfluidic device (scale bar − 10 μm). (Top right) output image denoised by NIDDL (scale bar − 10 μm). (Middle left) single pixel neuron activities extracted

from the noisy video for worm 2 in the above image. (Middle right) mean of ROI averaged neuron activities extracted from noisy video. (Bottom left) single pixel neuron activities extracted from the denoised video. (Bottom right) ROI averaged neuron activities extracted from the denoised video. Arrows indicate examples of activity transients lost when averaging across an ROI in noisy video. In contrast, clear activity transients are present in both single pixel and ROI pixel traces extracted from NIDDL denoised video. **D** (Top left) input noisy (low laser power) max-projection of an image stack from a large FOV video recording showing ventral cord motor neurons in a freely moving animal (scale bar − 10 μm). (Bottom left) output image deep-denoised by NIDDL (scale bar − 10 μm). (Top right) single pixel neuron activities extracted from the noisy video. (Bottom right) corresponding single-pixel neuron activities extracted from the deep-denoised video (arrows indicating coordinated activities). All data in **A**−**D** were collected on strain OH16230 using a 20×, 0.75 NA objective (online methods).

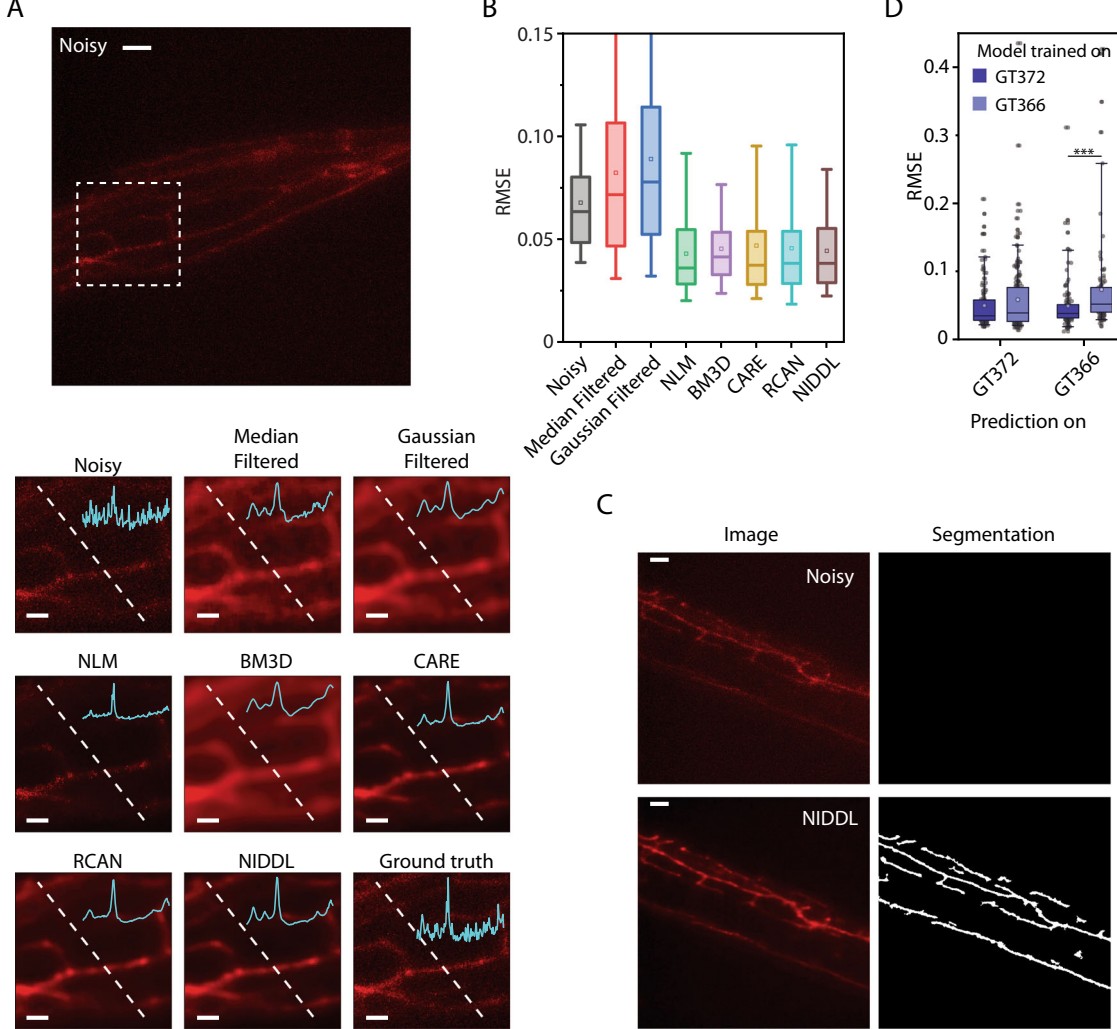

**Fig. 6 | NIDDL recovers complex neurite morphology of mechanosensory neurons. A** (Top) max projection of a noisy image stack (acquired at low laser power) showing neurites of harsh touch mechanosensory neuron PVD labeled with mScarlet (scale bar − 10 μm). Data comes from strain GT366. (Bottom) denoised outputs generated by various methods shown for dotted box in top image. Cyan trace in inset denotes pixel intensities along the dotted line (scale bar − 5 μm). **B** Comparison of RMSE accuracy across noisy images, and denoised images output by various methods. Data comes from strains GT372 and GT366. n = 86, 86, 86, 86, 86, 86, 443, 443 images across conditions. Box center indicates median, box edges 25th and 75th percentile, and whiskers indicate 5th and 95th percentile. Source data are provided in Source Data file. **C** NIDDL denoising of images facilitate neurite

segmentation. (Top) example noisy image showing harsh touch mechanosensory neuron PVD's neurites, no regions are detected in noisy images with simple morphological operations (see online "Methods" – 'Neurite segmentation'). (Bottom) corresponding NIDDL denoised output and segmented neurites in denoised image (scale bar − 10 μm). Data come from strain GT366. **D** Deep denoising RMSE accuracy comparison on noisy images from 2 strains (GT372 and GT366) that label neurites of gentle touch and harsh touch mechanosensory neurons respectively, when models are trained on strain-specific data. n = 129, 203, 118, 97 images for 4 conditions, ***p < 0.001, two-sided Holm-Bonferroni paired comparison. Box center indicates median, box edges 25th and 75th percentile, and whiskers 5th and 95th percentile. Source data are provided in Source Data file.

## Table 1 | List of strains used in this work

| Name | Genotype | Experiments | Reference |
|------|----------|-------------|-----------|
| ZIM504 | mzmEx199[Punc-31::NLSGCaMP5K; Punc-122::gfp]; lite-1 (xu7) | Used for (1) collecting training data for whole-brain images, (2) test denoising accuracy across imaging sessions (days), (3) test denoising accuracy across SNR in images (laser power settings), and (4) demonstrate high quality neuron activity trace extraction by denoising noisy whole-brain videos | [3,11] |
| OH16230 | otIs672 [rab-3::NLS::GCaMP6s + arrd-4:NLS:::GCaMP6s]. otIs670 provides a healthier alternative to otIs669, performing better in a variety of phenotypic assays. otIs670 [UPN::NLS::TagRFP-T + acr-5::NLS::mTagBFP2::H2B + flp-1::NLS::mTagBFP2::H2B + flp-6::NLS::mTagBFP2::H2B + flp-18::NLS::mTagBFP2::H2B + flp-19::NLS::mTagBFP2::H2B + flp-26::NLS::mTagBFP2::H2B + gcy-18::NLS::mTagBFP2::H2B + ggr-3::NLS::mTagBFP2::H2B + lim-4::NLS::mTagBFP2::H2B + pdfr-1::NLS::mTagBFP2::H2B + srab-20::NLS::mTagBFP2::H2B + unc-25::NLS::mTagBFP2::H2B + cho-1::NLS::CyOFP1::H2B + flp-13::NLS::CyOFP1::H2B + flp-20::NLS::CyOFP1::H2B + gcy-36::NLS::CyOFP1::H2B + gpa-1::NLS::CyOFP1::H2B + nlp-12::NLS::CyOFP1::H2B + nmr-1::NLS::CyOFP1::H2B + ocr-1::NLS::CyOFP1::H2B + osm-9::NLS::CyOFP1::H2B + srh-79::NLS::CyOFP1::H2B + sri-1::NLS::CyOFP1::H2B + srsx-3::NLS::CyOFP1::H2B + unc-8::NLS::CyOFP1::H2B + acr-2::NLS::mNeptune2.5 + ceh-2::NLS::mNeptune2.5 + dat-1::NLS::mNeptune2.5 + dhc-3::NLS::mNeptune2.5 + eat-4::NLS::mNeptune2.5 + flp-3::NLS::mNeptune2.5 + gcy-35::NLS::mNeptune2.5 + glr-1::NLS::mNeptune2.5 + gcy-21::NLS::CyOFP1::H2B:T2A::NLS::mTagBFP2::H2B + klp-6::NLS::mNeptune2.5::-T2A::NLS::CyOFP1::H2B + lim-6::NLS::mNeptune2.5::T2A::NLS::CyOFP1::H2B + mbr-1::NLS::mNeptune2.5::T2A::NLS::mTagBFP2::H2B + mec-3::NLS::CyOFP1::H2B::-T2A::NLS::mTagBFP2::H2B + odr-1::NLS::mNeptune2.5::T2A::NLS::mTagBFP2::H2B + srab-20::NLS::mNeptune2.5::T2A::NLS::mTagBFP2::H2B] V | Used for (1) collecting training data for whole-brain images, and ventral cord motor neurons, (2) test denoising accuracy across strains, (3) demonstrate high quality neuron activity trace extraction by denoising noisy videos of ventral cord neurons in restrained and freely moving animals | [68] |
| GT372 | aSi31[lox2272 Cbr-unc-119(+) lox2272 + mec-7p::GCaMP7F::ras-2CAAX::SL2::mScarlet-I::ras-2CAAX] II; unc-119(ed3) III | Used for (1) collecting training data for gentle touch neurons' neurite denoising, (2) demonstrate neurite denoising and segmentation | This work |
| GT366 | unc-119(ed3) III; aEx45[ser-2p3b::GCaMP7F::ras-2CAAX::SL2::mScarlet-I::ras-2CAAX + pDSP2(Cbr-unc-119(+)] | Used for (1) collecting training data for harsh touch neuron PVD neurite denoising, (2) demonstrate neurite denoising and segmentation | This work |

were collected for strain ZIM504 on different days using the same strategy. In this case, all datasets were collected at same laser power setting. To quantify prediction generalizability across image SNR levels, additional datasets were acquired using ZIM504 strain at very low and intermediate low laser power levels. To quantify prediction generalizability accuracy across strains with nuclear localized markers, data collected across two strains, OH16230 (nuclear localized GCaMP6s and TagRFP-T expression in all neurons) and ZIM504 ((nuclear localized GCaMP5K expression in all neurons), were used. In this case, data for all strains was collected at same laser power settings.

2. Ventral cord neurons data - Images of ventral cord motor neurons were collected using strain OH16230. Animals were synchronized to L4 stage and were immobilized in a microfluidic array device. 3D stacks (40 $z$ planes with 1 μm spacing) were collected at 10 ms exposure time, using SPlan Fluor ELWD 0.45 NA, 20× air objective. Two stacks were acquired for each animal, one at low laser power and one at the highest laser power setting available in microscope.

3. Neurite data - Images of neurites were collected using strain GT372 and GT366. These strains label different cells with different neurite morphology. GT372 labels gentle touch cells neurites that are sparser compared to harsh touch neuron PVD's neurites labeled in GT366. Animals were synchronized to L4 stage and were immobilized in a microfluidic array device. 3D stacks (40 $z$ planes with 1 μm spacing) were collected at 10 ms exposure time, using Plan Fluor 0.75 NA 40× air objective. Two stacks were acquired for each animal, one at low laser power and one at the highest laser power setting available in microscope.

### Synthetic whole-brain image data generation

To generate synthetic image data across a range of SNR levels, 3D stacks (128 × 128 × 30 pixels) were generated. Cells were simulated as 3D Gaussian distributions. Cell positions (mean of Gaussian distributions), cell sizes (3D covariance matrices of Gaussian distributions), and cell intensities (max peak of Gaussian distributions) were randomly generated for 60 cells and 3D intensity profiles of all cells were added together to form the image stack. Intensity profile of the resultant image was scaled to a maximum photon count level to specify the peak signal in image. Six photon count levels (20, 50, 100, 200, 500, and 1000) were used. This image specified the ground-truth clean image. To generate the corresponding noisy image, two kinds of noises were added to the clean image, photon shot noise (no parameter needed as the noise depends on each pixel's intensity level) and readout noise (normally distributed with mean 0 and 1 variance).

### Semi synthetic whole-brain video data generation

To generate semi synthetic 4D (3D + t) calcium imaging video data, 3D stacks (512 × 512 × 30 pixels) were generated for 100 time points. Here again, cells were simulated as 3D Gaussian distributions (as described in Synthetic image data generation section). However, here cell positions (means of Gaussian distributions) were taken from OpenWorm[63] atlas to mimic cell configuration in *C. elegans* head. 130 cells were randomly selected from OpenWorm atlas and positions of only those cells were used for a specific video. This mimics the fact that typically in whole brain recordings, not all cells are imaged due to low fluorophore expression. Further, temporal intensities for each cell were specified using experimental whole-brain recording datasets published previously[11]. A 100 frame window was randomly selected from published data, cell traces within the selected window were extracted from the published data, and each cell in synthetic video was randomly assigned a trace from the selected chunk. Thus realistic experimental calcium traces were present in synthetic video for each cell. Next, intensities of all frames were scaled to a maximum photon count level (using the maximum and minimum pixel intensity across all frames) to specify the peak signal in video. Four photon count levels (100, 200,

500, and 1000) were used. This specified the ground-truth clean video. To generate the corresponding noisy video, two kinds of noises were added to each frame, photon shot noise (Poisson noise) and readout noise (normally distributed with mean 0 and 1 variance).

## Calcium imaging data collection

We demonstrate deep denoising framework's capability to extract high quality calcium traces from noisy videos for three applications.

1.  High magnification head ganglion functional imaging - Data was collected using ZIM504 strain. Animals were synchronized to L4 stage and were immobilized in a microfluidic array device. Video (3D + t) stacks (30 $z$ planes with 1 µm spacing, × time points) were acquired at 10 ms exposure time, using Plan Fluor 0.75 NA 40x air objective. Noisy (low SNR) frames were acquired at low laser power. For each noisy stack, a ground-truth (high SNR) stack was acquired alternatively. Thus, the two stacks were not completely synchronous, however the time difference between two stacks was very small (~100 ms) compared to the dynamics of calcium signal. We compared the traces extracted from noisy video after denoising it with deep neural network with the traces extracted from ground-truth video to ensure that deep denoising does not introduce artifacts in traces.

2.  Low magnification functional imaging of ventral cord neurons in device - Data was collected using OH16230 strain. Animals were synchronized to L4 stage and were immobilized in a microfluidic array device. Video (3D + t) stacks (40 $z$ planes with 1 µm spacing, × time points) were acquired with 10 ms exposure time and SPlan Fluor ELWD 0.45NA 20× air objective. All stacks were acquired at low laser power settings.

3.  Low magnification functional imaging of ventral cord in freely moving animals - Data was collected using OH16230 strain. Animals were synchronized to L4 stage and were sandwiched between two agar pads on two cover-slips before imaging. 3D stacks (20 $z$ planes with 1 µm spacing, × time points) were collected at 10 ms exposure time using Plan Apo Lambda 0.75NA 20× air objective. All stacks were acquired at low laser power settings. At 20× magnification, animals were tracked easily while imaging using $z$ stage $x$-$y$ controller and kept in the field of view.

## Network optimization

We experimented with UNet[17,28], Hourglass[29] and DFCAN[21] architectures given the past success of these networks shown in several biological image analysis tasks such as image restoration, pose prediction, segmentation etc. Architecture details of networks are shown in Supplementary Fig. 1, 2. We tested with three hyper-parameters and training settings as described below. In all cases, the network takes as input a noisy (low SNR) image ($512 \times 512 \times d$) and through applications of convolutional layers with non-linear activation (ReLU), max-pooling, up-sampling, feature concatenation or summation generates an output image ($512 \times 512 \times d$). Here $d$ is the depth of input and output images. We experimented with different $d$ values as described below. Parameters in networks were trained using stochastic gradient descent with AdamOptimizer (learning rate 0.001) such that the output image is as close as possible (per some loss function) to the corresponding clean (high SNR) image. Training was performed on computing clusters using 16GB or 32 GB GPUs.

1.  Architectures – The following CNN architectures were tested.
    i.  UNet – An architecture very similar to conventional UNet architecture was used with 4 down-sampling/max pooling and 4 up-sampling layers. In this case, the first feature map had 32 channels (i.e., $512 \times 512 \times 32$). Depth (number of channels) of feature maps after each max pooling based down-sampling doubled and depth of feature maps after each up-sampling layer halved. Similar to conventional

UNet, long range residual connections were included that concatenate feature maps in down sampling to the feature maps in up-sampling layers.

ii.  UNet_fixed – This architecture is the same as the Unet architecture. However in this case the depth of all feature maps was fixed to 32. Doing so significantly reduced the model size compared to Unet and decreased the network training and inference time without any decrease in accuracy (Fig. 1B and Supplementary Fig. 3).

iii.  Hourglass_wores – An architecture very similar to the conventional Hourglass architecture was used. Compared to the Unet architecture, where long range residual connections are a direct concatenation of feature maps in down-sampling layers to feature maps in up-sampling layers, Hourglass architecture has side blocks with trainable parameters (see Supplementary Fig. 2) that extract features from down-sampling layers before max-pooling them and adding them to the feature maps in up-sampling layers. This enables the network to extract relevant information from feature maps in down-sampling layers. In our implementation, different from conventional Hourglass architecture, depth of feature maps within each convolutional block was not kept fixed thus it was not possible to include short range residual connection within convolutional block as it requires summation of input and output with same feature depth. Depth of the first feature map was set as 32. Depth of feature maps after each down-sampling layer doubled and depth of feature maps after each up-sampling layer halved

iv.  Hourglass_wres – Architecture same as Hourglass_wores was used. However, in this case depth of feature maps in each layer was kept fixed as 32. Further, short range residual connection within each convolutional block was used. Keeping the depth of feature maps fixed to 32 significantly reduced the model size compared to Hourglass_wores, and decreased the network training and inference time without any decrease in accuracy (Fig. 1B and Supplementary Fig. 3).

v.  DFCAN – Architecture implementation was borrowed from previously published code[21]. We were not able to train the network when input size was $512 \times 512 \times 1$ with published architecture on 32GB memory GPU due to large GPU memory requirements. Thus, we reduced the feature depth in FCAB (feature channel attention blocks) to 32 compared to 64 in published implementation. Further, we trained the network with $128 \times 128 \times 1$ images instead of $512 \times 512 \times 1$ (used for previous methods in this section) to further reduce memory requirements. The number of RCABs (residual channel attention blocks) was kept as 4 and each RCAB contained 4 FCABs, same as published implementation. By design for our task, the output image size is the same as the input image size; thus, scale factor was set to 1. With these settings, we trained several instances of networks with random selection of same of amount of training data used for previous architectures. However, the network did not train well as the output images of the trained networks looked empty across all training instances. This could be due to not enough training data needed by DFCAN. Thus, DFCAN was not considered for further optimization.

All models were trained on the same set of training data and accuracy was tested on a separate held-out dataset consisting of 600 images. Based on comparable or higher accuracy achieved by UNet_fixed and Hourglass_wres architectures compared to other architectures and much smaller memory footprint of these

architectures, we selected these architectures for our applications. Small memory footprint also provides the benefit of faster training and faster inference, thus making models user friendly and enabling real time applications.

Apart from architecture type, we also sought to determine if larger filters in convolutional layers can increase accuracy as they can take into account longer range spatial context in images. To do so, we compared the prediction accuracy of the two selected architectures for two sizes of convolution filters ($3 \times 3$ vs $5 \times 5$). Since we did not see significant differences in accuracy when using $5 \times 5$ filters, and models with $3 \times 3$ filters have smaller memory footprint, we used $3 \times 3$ filters.

2. Loss function – Two kinds of loss functions have been used previously for image restoration tasks, L2 loss, and L1 loss[17]. We asked if one loss function may achieve higher denoising accuracy on some datasets whereas the other may achieve higher accuracy on others. Thus, we trained the networks with both loss functions and compared the accuracy of models across them for all datasets. For high-magnification head ganglion dataset, we found that the accuracy of all architectures was comparable across L2 and L1 losses, with L2 loss performing slightly better in SSIM metric (Supplementary Fig. 3). Further, L1 loss showed more stable training, as different rounds of training the network from scratch showed lower variability in accuracy. In comparison, L2 loss-trained network showed greater variability in performance across different rounds of training. For harsh and gentle touch mechanosensory neurons' neurite data, L2 loss performed slightly better than L1 loss (Supplementary Fig. 18).

3. 2D vs 2.5D vs 3D training – To identify if depth context in 3D image stacks can improve de-noising performance, we tested several models (Supplementary Fig. 4).

    a. 2D models that take as input 1 low SNR image ($512 \times 512 \times 1$) and output 1 high SNR image ($512 \times 512 \times 1$).

    b. 2.5D models that take as input a noisy 2D image and $d$ $z$-planes above it and below it ($512 \times 512 \times (2d + 1)$) and outputs 1 high SNR image ($512 \times 512 \times 1$). Thus the network uses contextual information in $z$ planes above and below the image to be de-noised. The output of the network corresponds to the center $z$-plane of the input, i.e., the loss is minimized with respect to the center $z$ plane. We tested two values of $d$ with $d = 1$, and $d = 2$. Higher values of $d$ increases the memory footprint of training.

    c. 3D models that take as input a 3D image stack consisting of $d$ $z$-planes and outputs a 3D stack consisting of $d$ $z$-planes. Thus, all $z$-planes in the 3D input stack are de-noised simultaneously. Here again we tested two values of $d$ with $d = 1$, and $d = 2$.

All models were trained on the same set of training data, and accuracy was tested on a separate held-out but same for all dataset consisting of 600 images. Across these models we found that 2D models performed best. In principle, taking contextual information into consideration could improve performances. Our observation can be explained by the following. For 2.5D models, noise in $z$-planes around the center $z$ plane confused the model to focus on denoising center $z$-plane; further, for 3D models we had to reduce the batch size while training due to memory constraints, which could reduce training performance.

### Denoising and extracting calcium traces
**Whole-brain video.** Low SNR video collected at low laser power was first de-noised using a network trained on whole-brain image dataset. The trained network takes as input individual noisy $z$ planes ($512 \times 512 \times 1$) of 3D image stacks in the video and outputs high SNR $z$ planes ($512 \times 512 \times 1$), which were subsequently combined to form the de-noised video. To obtain activity traces, nuclei in ground-truth video were first segmented using a Gaussian mixture based segmentation method. Segmented nuclei were tracked across frames using an automated tracking algorithm. Generated tracks of cells were manually inspected and tracks for cells with minor tracking errors were semi-manually corrected. Single pixel activity traces were extracted using the centers of the tracked segmented masks. Activity traces were also extracted by averaging intensity of voxels with an ROI ($5 \times 5 \times 3$ size). Figures captions indicate what type of activity traces have been shown in the figure. The same segmented masks were used to extract activity traces from the noisy video and the de-noised video as well to get consistent activity traces across videos and avoid any comparison artifacts due to differences in cell segmentation procedures across videos.

**Ventral cord motor neurons in device.** Here we de-noise maximum projection images of 3D stacks in the video instead of whole 3D stacks as in whole-brain video denoising case. Thus, in this case the trained network takes as input a maximum projection image of a noisy stack ($512 \times 512 \times 40$ converted to $512 \times 512 \times 1$) in the video and outputs high SNR maximum projection stack ($512 \times 512 \times 1$). Neuron activity traces were extracted from the maximum projection denoised output. Single pixel activity traces were extracted using the centers of the nuclei. Activity traces were also extracted by averaging intensity of pixels with an ROI ($3 \times 3$ size). Figures captions indicate what type of activity traces have been shown in the figure.

**Ventral cord motor neurons in freely moving animal.** Here again we de-noise maximum projection images of 3D stacks in the video instead of whole 3D stacks. The trained network takes as input a maximum projection image of a noisy stack ($512 \times 512 \times 20$ converted to $512 \times 512 \times 1$) in the video; and outputs high SNR maximum projection stack ($512 \times 512 \times 1$). Neuron activity traces were extracted from the maximum projection de-noised output. To do so, cells in maximum projection images were tracked manually using Manual-Tracking plugin in Fiji. Subsequently, single pixel activity traces were extracted from both noisy and de-noised videos using track centers.

### Accuracy quantification

1. Image denoising accuracy – We quantify image denoising accuracy using 3 metrics – root mean square error (RMSE), peak signal to noise ratio (PSNR) and Structural Similarity Index (SSIM)[64]. For each of these metrics, high SNR (ground-truth) image was taken as the reference, and corresponding low SNR (noisy) and network output (denoised) images were compared to the reference. Since maximum intensity value or dynamic range of low SNR (noisy) images is much lower than those in high SNR (clean) images, we normalized intensity values in all images first before calculating the accuracy metrics to prevent arbitrary inflation of errors. Same methodology was used for network optimization and accuracy analysis across all datasets including high magnification whole-brain dataset, low magnification ventral cord imaging, and high magnification neurite dataset).

2. Activity trace from experimental whole-brain video – Single pixel neuron activity traces were extracted from the noisy video, ground-truth video and deep denoised video (see "Methods" – 'Calcium imaging data collection for video acquisition details' and "Methods" – 'Denoising and extracting calcium traces for activity extraction details'). Accuracy was quantified by

    a. Comparing MAE (mean absolute error) of traces extracted from noisy and denoised videos to the traces extracted from

ground-truth video.

$$\text{MAE}_{\text{noisy}} = \frac{1}{T}\sum_{t=1}^{T}|y_{\text{noisy},t} - y_{gt,t}| \qquad (1)$$

$$\text{MAE}_{\text{denoised}} = \frac{1}{T}\sum_{t=1}^{T}|y_{\text{denoised},t} - y_{gt,t}| \qquad (2)$$

Here $T$ denotes the number of time-points in time-series, $y_{\text{noise},t}$ and $y_{\text{denoised},t}$ denote neuron activities extracted from noisy video and denoised videos respectively at time $t$, $y_{gt,t}$ denote neuron activity extracted from ground-truth video at time $t$.

b. Comparing Pearson correlation coefficient of neuron activity traces extracted from noisy and denoised videos to the traces extracted from the ground-truth video.

## Neuron activity - curvature correlation in freely moving animal

To calculate the curvature of the body as *C. elegans* moves, a 4th degree polynomial was fitted to the coordinates of tracked ventral cord neurons to get ventral cord backbone. Since some cells go out of field of view during animal motion, cells that were consistently present across all frames were used to extract a backbone chunk and curvature analysis was performed using this backbone chuck only. The backbone chunk was divided into 100 segments (sampled at 100 points) and tangent angles to the backbone were calculated at these points. Neuron activity traces were cross-correlated to tangent angles at all points (shown in heat maps in Supplementary Fig. 17D). To quantify the improvement in neuron activity-curvature cross-correlation in deep denoised videos, cell traces were cross-correlated to local tangent angles i.e., tangent angles to the backbone at cell's location, and maximum absolute value of the cross-correlation across cells was compared when activity traces were extracted from noisy videos or denoised videos.

## Neurite segmentation

Harsh touch neuron PVD's neurites were segmented in noisy and deep denoised images using custom script in MATLAB. The custom script included basic operations with functionalities available in MATLAB – (1) image was sharpened (2) binarized with adaptive thresholding, (3) morphologically eroded to remove segmented noise (4) small holes were filled in image complement, and (5) structures smaller than fixed pixel size were removed.

## Comparisons against other methods

We compared the denoising performance of our optimized architectures with several methods across three accuracy metrics; RMSE, SSIM, PSNR. The methods included traditional methods such as Median Filtering and Gaussian Filtering, advanced non-deep learning based methods such as NLM[65], BM3D[66,67], and deep learning based methods such as CARE[17], and RCAN[19]. Below we provide implementation details of these methods.

1. Median Filtering was implemented using default MATLAB function. Three window sizes (3, 5, and 7) for filtering were tried for each dataset and results were reported for best performing window size.
2. Gaussian Filtering was implemented using default MATLAB function. Three kernel sizes or standard deviation values (1, 3, and 5) were tried for each dataset and results were reported for best performing window size.
3. NLM method was implemented using default MATLAB function. No parameters were set for NLM method as it automatically estimates the degree of smoothing based on standard deviation of noise in the image.
4. BM3D method was implemented using MATLAB implementation available here https://webpages.tuni.fi/foi/GCF-BM3D/. Four

different values of noise standard deviation were tried (0.05, 0.1, 0.2, and 0.5) and results were reported for best performing value for each data set.

5. CARE was implemented using the code provided at https://github.com/CSBDeep/CSBDeep. The network was trained using the same data as NIDDL, i.e., paired low SNR and high SNR images. Default parameters provided in code were used for training except unet_n_depth was set as 4 to be comparable to vanilla UNet architecture that we tried.
6. RCAN was implemented using code provided at https://github.com/AiviaCommunity/3D-RCAN. The network was trained using the same data as NIDDL, i.e., paired low SNR and high SNR images. Default parameters set in code were used for training.

## Inference runtime comparisons - system configuration

To compare inference runtime across various deep learning methods, the following system configurations were used.

1. GPU – Quadro M4000, memoryClockRate(GHz): 0.7725, compute capability: 5.2, totalMemory: 8.00 GiB.
2. CPU – Intel® Xeon® CPU E5-1620 v4 @ 3.50 GHz, RAM: 32 GB, 64-bit Operating System, x64-based processor.

## Statistical tests

All statistical comparisons were performed using the Paired Comparison Toolbox in Origin. Holm–Bonferroni paired test was used.

## Reporting summary

Further information on research design is available in the Nature Research Reporting Summary linked to this article.

# Data availability

Sample datasets to run trained networks are available at https://github.com/shiveshc/whole-brain_DeepDenoising. Raw imaging datasets used to train networks and various calcium imaging datasets will be available upon request due to the large size. The data generated in this study are provided in the Source Data file. OpenWorm atlas was obtained from https://doi.org/10.1098/rstb.2017.0382[63]. Source data are provided with this paper.

# Code availability

Code with example datasets is available at https://github.com/shiveshc/whole-brain_DeepDenoising. Instructions on how to run code on sample datasets and train on new datasets are available in the same repository.

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

## Acknowledgements

The authors thank the Zimmer Lab for providing ZIM504 strain and the Hobert Lab for providing NeuroPAL strains. The authors acknowledge the funding support of the U.S. NIH (R01NS096581, R01MH130064, and R01NS115484) and the U.S. NSF (1764406 and 1707401) to H.L. Some nematode strains used in this work were provided by the Caenorhabditis Genetics Center (CGC), which is funded by the NIH (P40 OD010440), National Center for Research Resources and the International *C. elegans* Knockout Consortium. This research was supported in part through research cyberinfrastructure resources and services provided by the Partnership for an Advanced Computing Environment (PACE) at the Georgia Institute of Technology, Atlanta, Georgia, USA. The authors declare no competing interest.

## Author contributions

S.C. and H.L. designed the algorithm, experiments, and methods. S.C. collected training data for whole-brain imaging, ventral cord motor neurons imaging and mechanosensory neurites imaging. S.M. collected training data for whole-brain imaging. S.C. collected functional imaging videos. S.C. and H.L. analyzed the algorithm performance and data. S.C., S.M., and H.L. wrote the paper.

## Competing interests

The authors declare no competing interests.
