## [Peer Review File · Nature Communications]

Fast, Efficient, and Accurate Neuro-Imaging Denoising via Supervised Deep LearningReviewers' comments:

Reviewer #1 (Remarks to the Author):

This manuscript describes a SNR improvement method for volumetric functional imaging based on an optimized convolutional neural network. The authors demonstrated the strong generalization ability of the network model on various applications, including whole-brain, motor neurons imaging in freely-moving animals. The major advantage of this work over the alternative approaches is the optimized CNN model which has achieved faster training and inference speed with similar performance and fidelity. Meanwhile, the weakness of this work lies in the fact that both the main framework of the network (Weigert M., et al., Nature Methods, 2018, 15, 1090) and the concept to use deep-learning strategy to improve the SNR of calcium signals (Li, X., Zhang, G., Wu, J. et al., 2021) have been reported in previous works. While this lightweight, data-efficient network is believed to show great potentials for high-speed and long-term calcium imaging in many biological assays, I still have some major questions that may help to improve the quality of this manuscript.

1. Besides less computational consumption, how are the SNR improvement and reconstruction accuracy as compared to the state-of-the-art methods, such DSP-Net and 3D RCAN?
2. The authors show the performance comparison between the proposed network and other training modes in Supplementary Figure 4 and claim that the network trained with 2D images performed best based on 2D evaluation criteria. However, given that the signals usually distribute in three dimensions at different depths, the reconstruction accuracy in z dimension is also important. Therefore, more performance comparisons in true 3D should be made in this manuscript.
3. Why the performance of the network model is sensitive to the laser power? Will this issue be solved by using more complex network model or improved training strategy? Besides, what is the acceptable degree of laser intensity fluctuation?
4. The authors set the exposure time to 10 ms at each plane in all experiments. However, in the freely-moving animal experiments shown in supplementary video 4, there are still noticeable motion blurs, which obviously induce reconstruction artifacts. It should be noted that a higher temporal resolution for live volumetric imaging has been achieved in previous works (Wu, Y. et al., Nature Methods, 2019, 16, 1323; Wang, Z. et al., Nature Methods, 2021, 18, 551; Wagner, N. et al., Nature Methods, 2021, 18, 557) and they are not appropriately introduced in this work yet. I'm curious that whether the exposure time can be further reduced, and how the accuracy of the network inference will be affected under this circumstance?

Minor comments:

1. In supplementary video 3, while there are no signals in the raw frames, the network still generates output signals. Similar problem is also shown in supplementary video 4.
2. In figure 2g, the RMSE values of CNN have a larger dynamic range as compared to the traditional denoising approaches. What is the reason for this?
3. The authors should indicate the network architectures used in each experiment. Also, the model name UNet_fixed or the Hourglass_wres is very awkward-sounding. A succinct name would be much more favorable.

Reviewer #2 (Remarks to the Author):

Summary:

The authors use supervised learning to train multiple convolutional neural networks (CNNs) for the denoising of fluorescence microscopy images.

Training data (including ground truth) is collected by recording pairs of subsequent high-exposure and low-exposure images.

The authors compare the results obtained by the different architectures and training modalities against each other and against basic denoising baselines.

They find that their method outperforms the baselines using their established evaluation metrics and make some interesting observations regarding which hyper-parameters and architectures should be used.

The authors use their denoised images for downstream analysis and explore the merits of denoising for neuron activity tracing.

Strengths:

- I appreciate the work the authors did on the effects of denoising on downstream analysis of their denoised data.
- I appreciate the extensive experiments the authors did regarding their network architectures, training conditions, generalisation, and hyper-parameters.
- The method and the experiments are presented in a thorough and easily understandable way. The figures look at the data and results from various angles and discuss all aspects I can think of.
- The approach and methods used by the authors are sensible.

Weaknesses:

- The denoising approach the authors use is in itself not novel.

The paper is a thoroughly done application and evaluation of a well established method, supervised CNN-based denoising.

There are by now many software tools available that can achieve this type of denoising for microscopy.

On the other hand, the authors showcase the possibilities of this method for specific kinds of downstream analysis, which is valuable.

There also is merit in the experiments done using different network architectures, loss-functions, etc..

- The baseline methods (Gaussian blur, and Median filter) chosen by the authors are very weak. There are multiple generations of superior non-deep-learning-based methods (e.g., Non-Local Means, or BM3D) available.

Showing that deep learning is superior to these types of methods is, I believe, generally accepted.

- In their introduction, the authors write that their network can be "trained with as few as 500 single images that are temporally independent and collected across different samples."

I found the term (temporally) "independent" (the term is used multiple times throughout the paper) prone to create a misunderstanding. I read this as meaning the training pairs do not have to show the same content. This would be a really important contribution.

However, as can be seen in Figure 1, the images pairs do have to correspond. Which means the setup is well established.

- I do not think the argument made by the authors regarding unsupervised learning is very convincing. They write that unsupervised methods require signals to be "similar in adjacent frames; moving samples would require non-trivial pre-registration of the frames; further, a large set of training data

would be needed to achieve generalizability."

I am afraid I disagree with these points.

As I see it, (i) unsupervised (or self-supervised) methods do not require images in adjacent frames to be similar.

They can be applied to individual images in the same way as the method presented by the authors and would not require pre-registration.

Also, (ii) Unsupervised methods do not require the recording of training data. The fact that they can cope without ground truth makes it possible to train on the very data that is to be denoised.

This means that training data is by definition available and the ability to generalise is arguably far less important.

All in all, I do not think that the authors have to argue against unsupervised learning and for supervised learning here. After-all, supervised learning is the gold standard regarding quality and well established.

Conclusion:

All-in-all my main concern is the novelty and the weak baselines.

Still, I think the paper has merits. I have to admit that I am not completely confident in my judgement on how valuable the insights here are for the community.

I wonder if the paper could be rewritten with a different pitch to focus even more on the benefits for downstream analysis and on the insights gained about training and hyperparameters in a real life setting, acknowledging that the approach itself is already established to achieve state-of-the-art results.

Reviewer #3 (Remarks to the Author):

This paper presents a supervised deep learning approach to denoise fluorescence microscopy images. In recent years a variety of deep neural networks for resolution enhancement have been proposed such as CARE [1], SRResNET [2], ESRGAN [3], and RCAN [4]. Specially CARE, which also uses 'ground-truth' data and a UNet architecture, seems to be very similar to this work. The main contribution of this paper is to use a UNet or Hourglass architecture but with fewer channels, which reduces memory footprint and provides the benefit of faster training and inference, apparently without impacting accuracy.

Major concerns:

Weak baselines: The accuracy of the denoising CNN is only compared to median filtering and Gaussian filtering, but not other state-of-the-art neural networks, such as CARE or RCAN.

To show that the proposed CNN still achieves state-of-the-art accuracy, those comparisons should be added in Fig 1F and 2G&J as well as supplementary Figure 8,12,13,... instead of merely delegating it to Supplementary Figure 3 where the UNet might actually be CARE.

Bad signal extraction method: Using single pixel activity is a very unusual crude method to extract neuronal calcium traces. Please use a method employed in practice such as CNMF [5] which is implemented in the popular pipelines Suite2P [6] and CalmAn [7].

I also suggest changing the title of the paper, because it is currently misleading. The output of the CNN is a denoised video, not neuronal calcium activity traces, as the title seems to imply. To recover the latter the denoised video would be fed into one of the mentioned pipelines.

Clarity: It has been unclear to me, already in the abstract, where you see the main target application. On one side you state your method is "obviating ultrafast imaging", on the other "the framework will

enable faster imaging". Which one is it? It seems one could just use CARE for slow imaging and e.g. Deepinterpolation [8] or DeepCAD [9] for fast imaging, maybe speeding up the inference for the latter by using your idea of fixed channel depths.

You claim an advantage of supervised learning is to train with temporally independent data, but as far as I understand this is not really the case because the signal in your high-SNR ground truth data is highly correlated with the corresponding low-SNR data (time difference ~100ms).

Minor:

The reference to your GitHub repo is neither in the main manuscript nor the supplement.

Installation did not work for me on Linux. Ok, you didn't promise that it would, but at least in my experience at various academic institutions, they all use Linux for scientific computing and it would be in your favor to support it.

Installation on my MacBook Pro worked, but the demo failed noting an illegal hardware instruction, possibly due to the M1 chip or the missing GPU.

Due to lack of access to Windows, I couldn't test your OS of choice.

page 2:

mischaracterization of unsupervised learning. DeepInterpolation considered volumetric fMRI data, and like them you yourself process 3D data by independently processing 2D z-planes. Similar signals in adjacent frames is not an assumption of Noise2void.

page 3:

If the "models can be used without expensive GPU", does one need no GPU at all? Or merely a cheap one?

Missing comparison to CARE or RCAN

Missing reference to Fig 1J when referring to Supplementary Figure 12.

page 4:

bad signal extraction method, missing references to proper methods

Denosing does not recover the correlation structure of the ground truth traces, but seems to introduce or exaggerate correlations. The ground truth is far less but still noisy. I suggest performing the analyses of correlations also on the semi synthetic imaging data.

page 5:

Not all unsupervised methods rely on multiple images of each sample with correlated signals, Noise2void does not.

supervised learning frameworks DO make assumptions on signals in the data: they rely on two images of each sample with correlated signals, a low-SNR and a high-SNR one.

You claim enabling faster frame rates was previously not possible. Faster frame rates are already possible using one of the other denoising nets [1-4]; denoising the obtained recording offline just takes longer due to slower inference. This is inconvenient but only a true problem if one would want to perform online real time analysis.

page 7:

I would cite the final journal version instead of the preprint for DeepInterpolation.

page 17:

Repeating myself, please use a state-of-the-art method for extracting calcium traces.

References for the Gaussian mixture based segmentation and the tracking algorithm are missing.

Figure 1:

caption: Your SL method only denoises the video; the next step in the pipeline is to use a method (usually CNMF) to extract high-quality Ca traces from this denoised video.

You don't show that your method outperforms existing methods like CARE or RCAN.

C: Which GPU was used?

How does this Figure look for unet, hourglass_wores, CARE and RCAN?

How does it look "without an expensive GPU"?

D: How far do the whiskers in the boxplot extend? Min and max, $1.5 \cdot \text{IQR}$, a percentile, something else?

How does the figure look for the other nets? Do they require more images?

F: What are the dots?

Whiskers show what?

What was the size of the Gaussian and median filter?

Comparison to e.g. CARE and RCAN is missing.

G: shouldn't the model trained on strain 1 predict better on strain 1 than the model trained on strain 2, but does not?

G-I: Is there an effect of training data size? The values of n differ markedly.

J: Why suddenly switch from RMSE to nMSE? Which formula defines nMSE?

Figure 2:

caption: typo in "cell makers"

B, H, I: Please use state-of-the art for signal extraction.

G, J: add comparisons to CARE and RCAN.

Supplementary Figure 3:

Isn't RMSE the (square root of the) L2 objective? How does minimizing L1 yield a lower L2 than minimizing L2 directly?

Remaining Supplementary Figures:

Address my main concerns: comparisons to CARE & RCAN, signal extraction using CNMF.

[1] Weigert, M. et al. Content-aware image restoration: pushing the limits of fluorescence microscopy. *Nat Methods* 15, 1090–1097 (2018).

[2] Ledig, C. et al. Photo-realistic single image super-resolution using a generative adversarial network. In *IEEE Conference on Computer Vision and Pattern Recognition* 4681–4690 (2017).

[3] Wang, X. et al. ESRGAN: enhanced super-resolution generative adversarial networks. in *Computer Vision—ECCV 2018 Workshops* (eds Leal-Taixé, L. & Roth, S.) 63–79 (Springer, 2019).

- [4] Chen, J. et al. Three-dimensional residual channel attention networks denoise and sharpen fluorescence microscopy image volumes. *Nat Methods* 18, 678–687 (2021).
- [5] Pnevmatikakis, E. et al. Simultaneous denoising, deconvolution, and demixing of calcium imaging data. *Neuron* 89(2), 285-299 (2016).
- [6] Pachitariu, M. et al. Suite2p: beyond 10,000 neurons with standard two-photon microscopy. *bioRxiv* 061507 (2017)
- [7] Giovannucci, A. et al. CalmAn an open source tool for scalable calcium imaging data analysis. *Elife* 8, e38173 (2019).
- [8] Lecoq, J. et al. Removing independent noise in systems neuroscience data using DeepInterpolation. *Nat Methods* 18, 1401–1408 (2021).
- [9] Li, X. et al. Reinforcing neuron extraction and spike inference in calcium imaging using deep self-supervised denoising. *Nat Methods* 18, 1395–1400 (2021).

We thank the reviewers for their thoughtful read and bringing up various points. The reviewers converged on two main points: baseline comparison and advancement/significance of the work. We have done extensive revision, mainly to expand the text and highlight the information pointing to the advancement in our work. We now use an acronym to refer to our work for simplicity – NIDDL for Neuro-Imaging Denoising via Deep Learning.

Previous and new **direct benchmarking** with other methods show that NIDDL uses a very **small memory footprint**, highly **efficient** (very **fast inference speed** and **fast training time**), **without sacrificing accuracy**. This enables many labs to use the tool without expensive GPUs, and furthermore, the speed will allow applications with near real-time feedback, e.g. optogenetics control.

Our responses are in blue and indented. Reviewers' original comments are in black. To declutter the responses, we have separated the reviewers' minor comments to be at the latter half of the document. Note that because we greatly expanded the document and added more results, the figure references may be different from the original document.

Reviewer #1 (Remarks to the Author):

This manuscript describes a SNR improvement method for volumetric functional imaging based on an optimized convolutional neural network. The authors demonstrated the strong generalization ability of the network model on various applications, including whole-brain, motor neurons imaging in freely-moving animals. **The major advantage of this work over the alternative approaches is the optimized CNN model which has achieved faster training and inference speed with similar performance and fidelity.** Meanwhile, the weakness of this work lies in the fact that both the main framework of the network (Weigert M., et al., Nature Methods, 2018, 15, 1090) and the concept to use deep-learning strategy to improve the SNR of calcium signals (Li, X., Zhang, G., Wu, J. et al., 2021) have been reported in previous works. While this lightweight, data-efficient network is believed to show great potentials for high-speed and long-term calcium imaging in many biological assays, I still have some major questions that may help to improve the quality of this manuscript.

We thank the reviewer for recognizing the advantage of our approach and potential important applications.

While we agree with the reviewer that the framework of the network and the utility of deep learning to improve the SNR have been reported some previous and very recent works, there are marked differences to our approach and targeted applications. Very importantly, Li et al (2021) and Lecoq et al. (2020) use temporally linked data (i.e. video) to train the denoising network; this approach takes advantage of temporally preserved (and uncorrelated) information in the video to train the network. While useful, this approach requires a very large training data set, which is expensive (both in terms of labor and in terms of retraining when experimental conditions change), and is prohibitive for certain applications (e.g. closed loop feedback). Further, this approach requires sample to be stationary or pre-registration of images, which currently bars the applications in many

experimental systems (including *C. elegans*, *Drosophila*, zebrafish etc., where behavior recording is highly desired).

Our approach is different. **A priori, it is neither guaranteed nor obvious that networks trained with non-sequential data (as in our approach) would preserve the temporal structure in neuron activity when applied to videos.** In our approach, with the careful collection of the training data, we show that these temporally independently collected training data indeed can work. Additionally, we highlight several advantages of our framework such as order of magnitude smaller training data, generalizability, and real-time inference speeds. Further, our goal with this work is also to demonstrate the usefulness of the established image denoising approaches (currently applied for cultured cells) for other model systems with specific applications. We have highlighted this point in the revision.

We have clarified these points in the text (Introduction, Results, Discussion).

1. Besides less computational consumption, how are the SNR improvement and reconstruction accuracy as compared to the state-of-the-art methods, such DSP-Net and 3D RCAN?

The reviewer missed a comparison in the original manuscript. We already compared against 3D RCAN (mentioned in Methods section). In the revision, we have compared with RCAN implementation from (Chen et al., 2021). We have also compared with CARE (Weigert et al., 2018). We show that our framework (NIDDL) achieves comparable accuracy (shown in Figure 1, 2, 5, 6 and associated supplemental figures). We also show that in general NIDDL is much faster, and more memory efficient.

2. The authors show the performance comparison between the proposed network and other training modes in Supplementary Figure 4 and claim that the network trained with 2D images performed best based on 2D evaluation criteria. However, given that the signals usually distribute in three dimensions at different depths, the reconstruction accuracy in z dimension is also important. Therefore, more performance comparisons in true 3D should be made in this manuscript.

We apologize for the confusion in the original manuscript. For 3D mode of training, we already compared denoising performance in 3D images. We have clarified in the text (Paragraph 3, section – “Optimized deep neural networks for denoising images”).

3. Why the performance of the network model is sensitive to the laser power? Will this issue be solved by using more complex network model or improved training strategy? Besides, what is the acceptable degree of laser intensity fluctuation?

We agree with the reviewer that is an important and interesting point. We have extensively characterized why the model is more sensitive to laser power but not to other parameters. We have added new results (Figure 3F) and discussion in the text.

“We hypothesized that as long as a minimum requirement for SNR is met, NIDDL can produce efficient denoising, and that the corruption of the signal by noise beyond a certain threshold cannot

be rescued by denoising. Indeed, this notion is corroborated by the characterizations of the SNR in the actual experiments (Fig. 3F) where the SNR levels across laser powers vary vastly, those across strains vary less, and across independent experiments sessions have similar SNR levels. These results demonstrate that as long as the imaging experiments meet a minimum SNR threshold (~ 20), NIDDL can efficiently denoise. This points to the advantages of NIDDL, where training data sets can be gathered in a distributed manner and from varied conditions (including from different strains), which would greatly lower the barriers for use in practice.”

4. The authors set the exposure time to 10 ms at each plane in all experiments. However, in the freely-moving animal experiments shown in supplementary video 4, there are still noticeable motion blurs, which obviously induce reconstruction artifacts. It should be noted that a higher temporal resolution for live volumetric imaging has been achieved in previous works (Wu, Y. et al., Nature Methods, 2019, 16, 1323; Wang, Z. et al., Nature Methods, 2021, 18, 551; Wagner, N. et al., Nature Methods, 2021, 18, 557) and they are not appropriately introduced in this work yet. I’m curious that whether the exposure time can be further reduced, and how the accuracy of the network inference will be affected under this circumstance?

We agree with the reviewer that several deep learning based advancements have been made that improve imaging speeds desirable by researchers. We have now introduced these works in “Introduction” section. We suspect that NIDDL can be paired with these techniques to further prevent photobleaching or improve imaging speed by reducing exposure time e.g. a combined model or a pipeline of models can be developed that can first perform denoising and then perform virtual refocusing or light-field reconstruction. We have added this point in the “Discussion”.

Reviewer #2 (Remarks to the Author):

Summary:

The authors use supervised learning to train multiple convolutional neural networks (CNNs) for the denoising of fluorescence microscopy images. Training data (including ground truth) is collected by recording pairs of subsequent high-exposure and low-exposure images. The authors compare the results obtained by the different architectures and training modalities against each other and against basic denoising baselines. They find that their method outperforms the baselines using their established evaluation metrics and make some interesting observations regarding which hyper-parameters and architectures should be used. The authors use their denoised images for downstream analysis and explore the merits of denoising for neuron activity tracing.

Strengths:

- I appreciate the work the authors did on the effects of denoising on downstream analysis of their denoised data.

- I appreciate the extensive experiments the authors did regarding their network architectures, training conditions, generalisation, and hyper-parameters.

- The method and the experiments are presented in a thorough and easily understandable way. The figures look at the data and results from various angles and discuss all aspects I can think of.

- The approach and methods used by the authors are sensible.

We appreciate the reviewer's comments.

Weaknesses:

The denoising approach the authors use is in itself not novel. The paper is a thoroughly done application and evaluation of a well established method, supervised CNN-based denoising. There are by now many software tools available that can achieve this type of denoising for microscopy.

On the other hand, the authors showcase the possibilities of this method for specific kinds of downstream analysis, which is valuable. There also is merit in the experiments done using different network architectures, loss-functions, etc..

We agree with the merits that this reviewer identified – that until demonstrated, a method's utility is not guaranteed, and the scientific community will not benefit from such methods until successful applications have been shown. Specifically, while the denoising methods have demonstrated success, their application for recovering high quality neuron activity traces is demonstrated in few systems only, and not in model organisms like *C. elegans* (which is motile, among other challenges).

Further, as mentioned in the response to Reviewer 1's similar comment, ***a priori*, it is neither guaranteed nor obvious that networks trained with non-sequential data (as in our approach) would preserve the temporal structure in neuron activity when applied to videos.** Li et al (2021) and Lecoq et al. (2020) use temporally linked data (i.e. video) to train the denoising network; this approach takes advantage of temporally preserved (and uncorrelated) information in the video to train the network. While useful, this approach requires a very large training data set, which is expensive (both in terms of labor and in terms of retraining when experimental conditions change), and is prohibitive for certain applications (e.g. closed loop feedback). Further, this approach requires sample to be stationary or training data to be pre-registered, which currently bars the applications in many experimental systems (including *C. elegans*, *Drosophila*, zebrafish etc., where behavior recording is highly desired). In our approach, with the careful collection of the training data, we show that these temporally independently collected training data indeed can work. We highlight the advantages of denoising methods for *C. elegans*' and other model organisms' functional imaging community.

We have highlighted these points in the text (Introduction and Discussion).

- The baseline methods (Gaussian blur, and Median filter) chosen by the authors are very weak. There are multiple generations of superior non-deep-learning-based methods (e.g., Non-Local Means, or BM3D) available. Showing that deep learning is superior to these types of methods is, I believe, generally accepted.

In the revision, we have compared with Non-Local Means, BM3D, RCAN implementation from (Chen et al., 2021) and CARE (Weigert et al., 2018). We show that our framework (NIDDL) achieves much higher accuracy compared to NLM, and BM3D and comparable accuracy to deep-learning based methods (shown in Figure 1, 2, 5, 6 and associated supplemental figures). We also show that in general NIDDL has much faster inference speeds, and more memory efficient.

- In their introduction, the authors write that their network can be "trained with as few as 500 single images that are temporally independent and collected across different samples." I found the term (temporally) "independent" (the term is used multiple times throughout the paper) prone to create a misunderstanding. I read this as meaning the training pairs do not have to show the same content. This would be a really important contribution. However, as can be seen in Figure 1, the images pairs do have to correspond. Which means the setup is well established.

This is very much related to the point made above on video versus image training datasets. We clarified in the text what we mean by "temporally independent" in the 3rd paragraph in the "Introduction" and 2nd paragraph in the "Discussion". We want to make distinction with unsupervised methods (Lecoq et al., 2020; Li et al., 2021) that require temporally sequential (or video) data for training. As mentioned, this approach takes advantage of the information preserved in the video data. In comparison, supervised methods like the one used by us do not require temporally sequential samples. In our approach, with careful collection of training data, we show that non-sequential training data can work as well, which has not been shown before in the literature. *A priori*, this is not a guaranteed outcome. Further, non-sequential data used for training provides several advantages such as ease of collection without needing ultrafast imaging rates, and no pre-registration of images.

- I do not think the argument made by the authors regarding unsupervised learning is very convincing. They write that unsupervised methods require signals to be "similar in adjacent frames; moving samples would require non-trivial pre-registration of the frames; further, a large set of training data would be needed to achieve generalizability." I am afraid I disagree with these points. As I see it, (i) unsupervised (or self-supervised) methods do not require images in adjacent frames to be similar. They can be applied to individual images in the same way as the method presented by the authors and would not require pre-registration.

We agree that the original text did not provide clarity. Briefly, unsupervised methods like Noise2Noise, DeepCAD, DeepInterpolation (Lecoq et al., 2020; Lehtinen et al., 2018; Li et al., 2021) require images in adjacent frames or samples to be similar. For instance, for Noise2Noise, multiple

examples of same sample with independent Gaussian noise are needed for training. Further, the samples are ideally pre-registered. Similarly, consecutive frames in ultrafast neuronal recordings can be thought of as images of same samples with independent noise present in each frame (Li et al., 2021). Also, DeepCAD demonstrated in their paper that denoising accuracy decreases with decrease in temporal sampling rate of video. This implies that previous methods use temporal information and if the signals in nearby frames are temporally inconsistent then accuracy decreases. Similarly, demonstrations in DeepCAD and DeepInterpolation are on registered data (with minor deviations). While some unsupervised methods like Noise2Void (Krull et al., 2019) do not require adjacent frames, these methods have been shown to achieve lower accuracy and have not been applied to calcium imaging. We have clarified this in the Introduction and Discussion sections.

Also, (ii) Unsupervised methods do not require the recording of training data. The fact that they can cope without ground truth makes it possible to train on the very data that is to be denoised. This means that training data is by definition available and the ability to generalise is arguably far less important.

We agree with the reviewer that unsupervised methods do not require recording of training data. We have clarified this in the text. As pointed out by the reviewer, we further note that by training the network on same data to be denoised, the networks may not be generalizable and may require retraining for each video recording or experiment, which may be prohibitively expensive or at least inconvenient for certain applications.

All in all, I do not think that the authors have to argue against unsupervised learning and for supervised learning here. After-all, supervised learning is the gold standard regarding quality and well established.

The point is well taken. We revised the text to discuss the differences between supervised and unsupervised methods. We brought up unsupervised methods because for calcium imaging, the literature thus far is mainly using unsupervised methods, which work pretty well for imaging mouse cortex. We wanted to show that those methods are not practical for our and other applications, and thus a new method is needed. We believe the context is clarified in the revised text.

Conclusion:

All-in-all my main concern is the novelty and the weak baselines.

Still, I think the paper has merits. I have to admit that I am not completely confident in my judgement on how valuable the insights here are for the community.

I wonder if the paper could be rewritten with a different pitch to focus even more on the benefits for downstream analysis and on the insights gained about training and hyperparameters in a real life setting, acknowledging that the approach itself is already established to achieve state-of-the-art results.

We have performed comparisons against more non deep-learning based methods as suggested by the reviewer. We re-wrote several portions of the manuscript to 1) acknowledge that denoising

based methods are established, 2) highlight the benefits of deep denoising on downstream analysis for functional imaging in *C. elegans* and other model systems, mainly the ability to gather training data in a distributed manner, to cheaply train and use the model to denoise, and to enable real-time feedback control during experiments.

Reviewer #3 (Remarks to the Author):

This paper presents a supervised deep learning approach to denoise fluorescence microscopy images. In recent years a variety of deep neural networks for resolution enhancement have been proposed such as CARE [1], SRResNET [2], ESRGAN [3], and RCAN [4]. Specially CARE, which also uses 'ground-truth' data and a UNet architecture, seems to be very similar to this work. The main contribution of this paper is to use a UNet or Hourglass architecture but with fewer channels, which reduces memory footprint and provides the benefit of faster training and inference, apparently without impacting accuracy.

Major concerns:

Weak baselines: The accuracy of the denoising CNN is only compared to median filtering and Gaussian filtering, but not other state-of-the-art neural networks, such as CARE or RCAN.

To show that the proposed CNN still achieves state-of-the-art accuracy, those comparisons should be added in Fig 1F and 2G&J as well as supplementary Figure 8,12,13,... instead of merely delegating it to Supplementary Figure 3 where the UNet might actually be CARE.

We thank the reviewer for pointing this out; we completely agree with the reviewer that we should benchmark with others and clarify the advantage of our techniques. We did do a comparison in the original manuscript (with RCAN), but perhaps not very obviously since it was buried in the supplemental info. In the revision, we added several additional methods and brought these comparisons to the main figures, as well as additional supplemental figures. For instance, we have compared with advanced non-deep learning methods such as NLN, BM3D and deep learning methods such as RCAN (Chen et al., 2021), and CARE (Weigert et al., 2018).

We show that our framework (NIDDL) achieves much higher accuracy compared to NLN, BM3D and comparable accuracy to CARE and RCAN (shown in Figure 1, 2, 5, 6 and associated supplemental figures). We also show that in general NIDDL has much faster inference speeds, are faster to train, and are more memory efficient. Therefore, the advantages are that we have the ability to gather training data in a distributed manner, to cheaply train and use the model to denoise, and to enable real-time feedback control during experiments.

Bad signal extraction method: Using single pixel activity is a very unusual crude method to extract neuronal calcium traces. Please use a method employed in practice such as CNMF [5] which is implemented in the popular pipelines Suite2P [6] and CalmAn [7].

We are happy to try other methods for neuronal calcium traces extraction as suggested by the reviewers. However, we note that these methods are not standard in *C. elegans* calcium imaging

community. Instead, traces are extracted by first segmenting the cells and then extracting single pixel or top brightest pixel traces (Ji et al., 2020; Kato et al., 2015; Larsch et al., 2013). Further, the methods mentioned by the reviewer require or perform pre-registration of images that may not be trivial for data from moving *C. elegans*.

I also suggest changing the title of the paper, because it is currently misleading. The output of the CNN is a denoised video, not neuronal calcium activity traces, as the title seems to imply. To recover the latter the denoised video would be fed into one of the mentioned pipelines.

Thank you for the comment. We have changed the title of the paper.

Clarity: It has been unclear to me, already in the abstract, where you see the main target application. On one side you state your method is "obviating ultrafast imaging", on the other "the framework will enable faster imaging". Which one is it? It seems one could just use CARE for slow imaging and e.g. Deepinterpolation [8] or DeepCAD [9] for fast imaging, maybe speeding up the inference for the latter by using your idea of fixed channel depths.

We revised the text. The original language is there any more. This is a related point to those from the other reviewers. We meant that in unsupervised methods such as (Lecoq et al., 2020) and (Li et al., 2021), calcium imaging datasets were acquired at fast imaging rates of 30Hz and 30Hz respectively. Further in (Li et al., 2021), decreasing performance of denoising was shown for temporally down-sampled data. While new microscopic techniques have been developed, 30Hz volumetric imaging rate is currently not possible in worm and fly functional imaging community, especially in the context of whole-brain imaging. Thus, with non-sequential pairs of (or non-video) data used for training in our method, ultra-fast imaging rate is not needed specifically for training data collection.

By "the framework will enable faster imaging speed" we make a separate point that denoising will enable faster imaging speed with use of low exposure time. This could help with temporal resolution, and also real-time feedback control scenarios.

We agree with the reviewer that one could just use CARE for slow imaging; however, *a priori*, it is neither guaranteed nor obvious that networks trained with non-sequential data (as in our approach) would preserve the temporal structure in neuron activity when applied to videos. Li et al (2021) and Lecoq et al. (2020) use temporally linked data (i.e. video) to train the denoising network; this approach takes advantage of temporally preserved (and uncorrelated) information in the video to train the network. While useful, this approach requires a very large training data set, which is expensive (both in terms of labor and in terms of retraining when experimental conditions change), and is prohibitive for certain applications (e.g. closed loop feedback). Further, this approach requires sample to be stationary, which currently bars the applications in many experimental systems (including *C. elegans*, *Drosophila*, zebrafish etc, where behavior recording is highly desired). In our approach, with the careful collection of the training data, we show that these temporally independently collected training data indeed can work. We highlight the advantages of denoising methods for *C. elegans*' and other model organisms' functional imaging community.

You claim an advantage of supervised learning is to train with temporally independent data, but as far as I understand this is not really the case because the signal in your high-SNR ground truth data is highly correlated with the corresponding low-SNR data (time difference $\sim 100\text{ms}$).

This is a similar point raised by Reviewer 2. We clarified in the text what we mean by “temporally independent data” (in introduction and discussion, as well as in the results section when appropriate). We agree with the reviewer that low-SNR and high-SNR images are highly correlated as with other supervised denoising methods. We apologize for the confusion. We want to make distinction with unsupervised methods (Lecoq et al., 2020; Li et al., 2021) that require temporally sequential or video data for training. As mentioned, this approach takes advantage of the information preserved in the video data. In comparison, supervised methods like the one used by us do not require temporally sequential samples. In our approach, with careful collection of training data, we show that non-sequential training data can work as well, which has not been shown before in the literature. *A priori*, this is not a guaranteed outcome.

Minor Comments from Reviewers:

Reviewer 1 Minor comments:

1. In supplementary video 3, while there are no signals in the raw frames, the network still generates output signals. Similar problem is also shown in supplementary video 4.

There are signals in raw frames but too noisy or dim to be seen by eye. We show these signals in figures 5C and 5D.

2. In figure 2g, the RMSE values of CNN have a larger dynamic range as compared to the traditional denoising approaches. What is the reason for this?

We have added discussions to interpret results. The dynamic range is similar to some other methods such as CARE.

3. The authors should indicate the network architectures used in each experiment. Also, the model name UNet_fixed or the Hourglass_wres is very awkward-sounding. A succinct name would be much more favorable.

We have renamed our approach NDDL and all figures are labeled with NDDL.

Reviewer 3 Minor Comments:

We believe these comments are all addressed.

The reference to your GitHub repo is neither in the main manuscript nor the supplement.

Installation did not work for me on Linux. Ok, you didn't promise that it would, but at least in my experience at various academic institutions, they all use Linux for scientific computing and it would be in your favor to support it.

Installation on my MacBook Pro worked, but the demo failed noting an illegal hardware instruction, possibly due to the M1 chip or the missing GPU.

Due to lack of access to Windows, I couldn't test your OS of choice.

We have referenced the GitHub repo in the methods.

page 2:

mischaracterization of unsupervised learning. DeepInterpolation considered volumetric fMRI data, and

like them you yourself process 3D data by independently processing 2D z-planes. Similar signals in adjacent frames is not an assumption of Noise2void.

We did include a thorough description of unsupervised methods (paragraph 3 in “Introduction” and paragraph 2 in “Discussion” sections) and clarify the missing details as mentioned by the reviewer.

page 3:

If the "models can be used without expensive GPU", does one need no GPU at all? Or merely a cheap one?

Missing comparison to CARE or RCAN

Missing reference to Fig 1J when referring to Supplementary Figure 12.

We have done new comparisons of inferences speed across our optimized architecture, CARE, and RCAN (Figure 1E) when inference is done with or without GPU. A commonly available GPU with 8GB memory was used (system configurations are mentioned in “Methods - Inference runtime comparisons system configuration” section). We describe the results in paragraph 3 “Optimized deep neural networks for denoising images” section.

We compared against RCAN originally (mentioned in Methods section); in the revision, we added many more characterizations and comparisons against other methods including CARE and RCAN (shown in Figures 1, 2, 5, 6 and associated supplement figures).

page 4:

bad signal extraction method, missing references to proper methods

Denoising does not recover the correlation structure of the ground truth traces, but seems to introduce or exaggerate correlations. The ground truth is far less but still noisy. I suggest performing the analyses of correlations also on the semi synthetic imaging data.

We note that methods like (Giovannucci et al., 2019; Pachitariu et al., 2016) while very popular in mice community **are not standard in the *C. elegans* community**. Instead in *C. elegans* community, signals are extracted from cell segmentations using brightest or top brightest pixels (Kato et al., 2015; Larsch et al., 2013). We have added text in results section to give background of signal extraction methods.

page 5:

Not all unsupervised methods rely on multiple images of each sample with correlated signals, Noise2void does not.

We updated the description on unsupervised methods.

supervised learning frameworks DO make assumptions on signals in the data: they rely on two images of each sample with correlated signals, a low-SNR and a high-SNR one.

We agree with the reviewer. We added clarification in the text in Discussion section.

You claim enabling faster frame rates was previously not possible. Faster frame rates are already possible using one of the other denoising nets [1-4]; denoising the obtained recording offline just takes longer due to slower inference. This is inconvenient but only a true problem if one would want to perform online real time analysis.

We agree with the reviewer that faster frame rates are possible with previous denoising methods. However, whether these methods can extract clean, temporally coherent calcium traces using non-sequential training data have not been shown. In our work, by demonstrating that these methods indeed work with temporally independent data, we argue for the adoption and benefit of these methods for worm, fly and other model systems community. We also note that with our optimized methods, online real time analysis could be achieved.

page 7:

I would cite the final journal version instead of the preprint for DeepInterpolation.

We updated the reference.

page 17:

Repeating myself, please use a state-of-the-art method for extracting calcium traces.

We note that methods mentioned by the reviewer (Giovannucci et al., 2019; Pachitariu et al., 2016) are not standard in *C. elegans* community. Instead, calcium traces are extracted first by segmenting cells and then extracting brightest or top brightest pixel traces.

References for the Gaussian mixture based segmentation and the tracking algorithm are missing.

These methods are in-house developed and currently unpublished. Some of these methods and their variants can be implemented easily by others, and we provide the results of the segmentation and tracking for this paper.

Figure 1:

caption: Your SL method only denoises the video; the next step in the pipeline is to use a method (usually CNMF) to extract high-quality Ca traces from this denoised video.

We highlight in Figure 1A that calcium signal extraction step is different than denoising video step.

You don't show that your method outperforms existing methods like CARE or RCAN.

We originally compared against RCAN (mentioned in Methods section). The revised manuscript contains many additional figures for comparison against CARE and RCAN among others.

C: Which GPU was used?

How does this Figure look for unet, hourglass_wores, CARE and RCAN?

How does it look "without an expensive GPU"?

We have added the requested information in "Methods – Inference runtime comparisons – system configuration" section. For GPU versus CPU, we have also added Figure 1E to clarify this point.

D: How far do the whiskers in the boxplot extend? Min and max, 1.5*IQR, a percentile, something else?

How does the figure look for the other nets? Do they require more images?

We have made appropriate changes to the figure legends describing meaning of the boxes and whiskers. Figures that compare NIDDL with other methods include appropriate markings such as whiskers and statistics.

F: What are the dots?

Whiskers show what?

What was the size of the Gaussian and median filter?

Comparison to e.g. CARE and RCAN is missing.

We have added description in figure legends to describe dots and whiskers. Sizes of Gaussian and median filters tried are mentioned in "Methods – Comparison against other methods" section. Briefly, several sizes for filters were tried and results were reported for best performing size according to SSIM metric. Comparisons to CARE and RCAN are added now in Figures 1, 2, 5, 6 and associated supplement figures.

G: shouldn't the model trained on strain 1 predict better on strain 1 than the model trained on strain 2, but does not?

The results show that RMSE is higher but SSIM is also higher for the model trained on strain 1 predicting strain 1 as compared to predicting strain 2.

G-I: Is there an effect of training data size? The values of n differ markedly.

Size of training data for each condition was higher than 600 images which is sufficient to train the networks shown in Figure 1G. N values indicate the test data size for each strain.

Further, we describe the effect of training data size in Figure 1C and supplementary Figure 6.

J: Why suddenly switch from RMSE to nMSE? Which formula defines nMSE?

We clarified in the text that the methodology was same for calculating RMSE and nMSE. We renamed nMSE to RMSE.

Figure 2:

caption: typo in "cell makers"

B, H, I: Please use state-of-the art for signal extraction.

G, J: add comparisons to CARE and RCAN.

We made corrections.

We note that we used methods that are standard in *C. elegans* community for signal extraction. We note that (Giovannucci et al., 2019; Pachitariu et al., 2016) are not standard in *C. elegans* community and may not be directly applicable to moving animal videos due to non-trivial pre-registration step.

Supplementary Figure 3:

Isn't RMSE the (square root of the) L2 objective? How does minimizing L1 yield a lower L2 than minimizing L2 directly?

Yes, RMSE is the square root of the L2 objective. We agree with the reviewer that this is indeed surprising. However, we noticed that network trained with L1 loss achieved more stable training and lower RMSE. This could be because L1 loss is more suitable to tackle the types of noise present in

experimental data. We have added discussion in paragraph 3 “Optimized deep neural networks for denoising images” section to interpret results.

Remaining Supplementary Figures:

Address my main concerns: comparisons to CARE & RCAN, signal extraction using CNMF.

We have compared against RCAN and CARE in many figures and supplemental figures as stated before.

We note that we used methods that are standard in *C. elegans* community for signal extraction. We note that (Giovannucci et al., 2019; Pachitariu et al., 2016) are not standard in *C. elegans* community.

- [1] Weigert, M. et al. Content-aware image restoration: pushing the limits of fluorescence microscopy. *Nat Methods* 15, 1090–1097 (2018).
- [2] Ledig, C. et al. Photo-realistic single image super-resolution using a generative adversarial network. In *IEEE Conference on Computer Vision and Pattern Recognition* 4681–4690 (2017).
- [3] Wang, X. et al. ESRGAN: enhanced super-resolution generative adversarial networks. in *Computer Vision—ECCV 2018 Workshops* (eds Leal-Taixé, L. & Roth, S.) 63–79 (Springer, 2019).
- [4] Chen, J. et al. Three-dimensional residual channel attention networks denoise and sharpen fluorescence microscopy image volumes. *Nat Methods* 18, 678–687 (2021).
- [5] Pnevmatikakis, E. et al. Simultaneous denoising, deconvolution, and demixing of calcium imaging data. *Neuron* 89(2), 285–299 (2016).
- [6] Pachitariu, M. et al. Suite2p: beyond 10,000 neurons with standard two-photon microscopy. *bioRxiv* 061507 (2017)
- [7] Giovannucci, A. et al. CalmAn an open source tool for scalable calcium imaging data analysis. *Elife* 8, e38173 (2019).
- [8] Lecoq, J. et al. Removing independent noise in systems neuroscience data using DeepInterpolation. *Nat Methods* 18, 1401–1408 (2021).
- [9] Li, X. et al. Reinforcing neuron extraction and spike inference in calcium imaging using deep self-supervised denoising. *Nat Methods* 18, 1395–1400 (2021).

Reviewers' comments:

Reviewer #1 (Remarks to the Author):

I appreciate the authors' point-to-point response and the substantive updates, which addressed most of my previous questions and comments on the performance of this approach. In the revised manuscript, the authors compared the proposed approach with contemporary DL-enhanced image restoration techniques. Given their approach is registration-free, the comparison has justified its technical advances over the alternative techniques. There is only a minor concern left: in the comparison with CARE and 3D-RCN, whether the training data are paired? using paired data for them would be more reasonable to me. In summary, this manuscript has been substantially improved through revision. When compared to current DL image augment approaches, its lightweight and user-friendly features make it unique for a few biological applications. I would be happy to recommend its publication at Nature Communications.

Reviewer #2 (Remarks to the Author):

Thank for the additional work and especially for including the additional baselines.

I am happy with the modified version and recommend publication.

Reviewer #3 (Remarks to the Author):

I haven't been aware that, as you correctly state, CNMF is not commonly used for imaging data from behaving *C. Elegans*. Indeed, Liam Paninski's group suggested dNMF for that specific purpose (Nejatbakhsh et al 2020).

However, you wrongly claim it was common practice to extract single pixel or top brightest pixel traces. The very papers you cite in support (Kato et al., 2015; Larsch et al., 2013) do NOT extract single pixel traces, but average over some pixels. (The center of the ROI might be the brightest pixel). Kato et al., 2015, average over 75 voxels, and Larsch et al., 2013, over a 4x4 or 6x6 pixel region. The researchers performing the first whole-brain calcium-imaging on freely moving worms (Venkatachalam et al, 2016; Nguyen et al, 2016) also computed mean pixel values to extract signals.

Please follow common practice and perform some local averaging when extracting signals.

Fig 4D

Denosing does not recover the correlation structure of the ground truth traces, but seems to introduce or exaggerate correlations. The ground truth is far less but still noisy. I suggest performing the analyses of correlations also on the semi synthetic imaging data.

Discussion, paragraph 2

Noise2Noise works well with only 2 images.

NIDLL also requires 2, i.e. multiple, images. Thus, your claim "supervised learning methods, do not rely on multiple images of each sample" is wrong.

In the intro you say "...can be trained with as few as 500 single images that are temporally independent and collected across different samples." I assume you mean 500 image pairs not single images? It should be made clearer that the training data consists of image pairs. The images within a pair are low/high SNR versions of the same clear image, whereas there is no temporal dependency between pairs.

We thank the reviewers for their suggestions and comments. We have responded to all the comments.

REVIEWER COMMENTS

Reviewer #1 (Remarks to the Author):

I appreciate the authors' point-to-point response and the substantive updates, which addressed most of my previous questions and comments on the performance of this approach. In the revised manuscript, the authors compared the proposed approach with contemporary DL-enhanced image restoration techniques. Given their approach is registration-free, the comparison has justified its technical advances over the alternative techniques. There is only a minor concern left: in the comparison with CARE and 3D-RCN, whether the training data are paired? using paired data for them would be more reasonable to me. In summary, this manuscript has been substantially improved through revision. When compared to current DL image augment approaches, its lightweight and user-friendly features make it unique for a few biological applications. I would be happy to recommend its publication at Nature Communications.

We thank the reviewer for their suggestions and comments. We note that in comparison with CARE and 3D-RCAN, the training data were same as that used by NIDDL, i.e. pairs of low SNR and high SNR images. We have made sure the text is clear on this in Online Methods – Comparison against other methods section.

Reviewer #2 (Remarks to the Author):

Thank for the additional work and especially for including the additional baselines. I am happy with the modified version and recommend publication.

We thank the reviewer for their suggestions and comments prior.

Reviewer #3 (Remarks to the Author):

I haven't been aware that, as you correctly state, CNMF is not commonly used for imaging data from behaving *C. Elegans*. Indeed, Liam Paninski's group suggested dNMF for that specific purpose (Nejatbakhsh et al 2020). However, you wrongly claim it was common practice to extract single pixel or top brightest pixel traces. The very papers you cite in support (Kato et al., 2015; Larsch et al., 2013) do NOT extract single pixel traces, but average over some pixels. (The center of the ROI might be the brightest pixel). Kato et al., 2015, average over 75 voxels, and Larsch et al., 2013, over a 4x4 or 6x6 pixel region. The researchers performing the first whole-brain calcium-imaging on freely moving worms (Venkatachalam et al, 2016; Nguyen et al, 2016) also computed mean pixel values to extract signals. Please follow common practice and perform some local averaging when extracting signals.

We agree with the reviewer that some previous works extract traces by averaging over pixels in ROI. We have rephrased the statement in the text. We also agree that averaging over many pixels is usually done to remove noise from neuron activity traces. We wish to note that while such strategy is used in practice, it requires the magnification and resolution in images are high enough so that each neuron is composed of multiple pixels/voxels e.g. 40X magnification whole-brain imaging. This is not always the case, especially in applications where large ROI is required, in which case, NIDDL has a significant advantage where single pixel/voxel denoising can recover the signal of interest. We elaborate on these points below with additional data.

As requested by the reviewer, we compared the traces extracted from whole-brain video dataset in Figure 4 using both single pixel and ROI (mean of 75 voxels in 5 x 5 x 3 window). Response Figure 1 below show the standard metrics (MAE

and correlation coefficient). ROI traces from denoised video achieve marginally higher accuracy compared to single pixel traces from the denoised video. Thus, high quality traces can be obtained with just one pixel by using NIDDL denoising. This capability of NIDDL has particular advantages in the next example (large ROI) that we show.

Response Figure 1. Comparison of activity extraction methods for whole-brain recording dataset in Figure 4. A) MAE and B) Pearson correlation coefficient of traces from noisy and denoised videos to traces from high SNR video. Single pixel traces from denoised video achieve similar accuracy (low MAE, high correlation coefficient) as ROI traces from denoised video.

Next, we extracted traces from noisy and denoised ventral cord motor neuron dataset (shown in Figure 5C and Supplementary Figure 16). These videos are at low magnification where each neuronal nucleus is represented by a few voxels. We show in Response Figure 2 below that while ROI traces (average of 9 pixels in 3 x 3 window) extracted from noisy video (column 2) have significantly reduced noise, many calcium activity bouts are not recovered; some examples are highlighted using arrows in Response Figure 2. In comparison, single pixel traces from denoised video (column 3) and ROI traces from denoised video (column 4) clearly recover such bouts (see examples in Response Figure 2). To highlight the advantage of denoising, we have added text in Paragraph 2 “High SNR calcium trace recovery using NIDDL” section. We have also made changes to Figure 5C and Supplementary Figure 16 to highlight these examples.

Response Figure 2. Additional examples of ventral cord neurons' activity traces extracted from in-device recordings. Four columns in each row show 1) single pixel neuron activity traces from noisy videos, 2) neuron activity traces from noisy videos extracted using ROI mean, 3) single pixel neuron activity traces from denoised video, and 4) neuron activity traces from noisy videos extracted using ROI mean for A) worm 1 in Figure 5C, B) worm 2, C) worm 3, D) worm 4. Arrows indicate examples of neuron activity transients that are lost when traces are extracted by averaging ROI pixels in noisy video. In comparison, single pixel traces or mean ROI traces extracted from denoising video recover such transients.

Fig 4D

Denoising does not recover the correlation structure of the ground truth traces, but seems to introduce or exaggerate correlations. The ground truth is far less but still noisy. I suggest performing the analyses of correlations also on the semi synthetic imaging data.

We performed the analysis on synthetic data as suggested (now included in Supplementary Figure 12C and E).

Correlational structure recovered by denoising is comparable to that in ground-truth video (Supplementary Figure 12B, C). Additionally, Pearson correlation coefficient of denoised traces to ground-truth traces is consistently higher than that achieved with traditional denoising methods (Supplementary Figure 12E).

We note that in whole-brain data (Figure 4), single pixel traces from high SNR ("ground-truth") video were obtained with still lower enough laser power to allow long enough recording. They are indeed less noisy but still noisy. Thus, the difference in correlational structure in whole-brain data (Figure 4) is likely because the "ground truth" not being exactly the ground truth. In comparison, ground-truth traces in synthetic videos are noise-free. Taken together, we believe the improvement in correlation structure from NIDDL processed data is not an exaggeration.

Discussion, paragraph 2

Noise2Noise works well with only 2 images.

NIDDL also requires 2, i.e. multiple, images. Thus, your claim "supervised learning methods, do not rely on multiple images of each sample" is wrong.

Thank you pointing out the confusion. We have corrected the statement in discussion section to highlight that NIDDL also requires 2 images per sample.

In the intro you say "...can be trained with as few as 500 single images that are temporally independent and collected across different samples." I assume you mean 500 image pairs not single images? It should be made clearer that the training data consists of image pairs. The images within a pair are low/high SNR versions of the same clear image, whereas there is no temporal dependency between pairs.

We have clarified this in the introduction.

Reviewers' comments:

Reviewer #3 (Remarks to the Author):

I am pleased with the revised version and recommend publication.